# Ice shelf basal channel shape determines channelized ice-ocean interactions

Chen Cheng [1] ✉, Adrian Jenkins [2,5], Paul R. Holland[3,5], Zhaomin Wang[1], Jihai Dong [4] & Chengyan Liu[1]

Growing evidence has confirmed the critical role played by basal channels beneath Antarctic ice shelves in both ice shelf stability and freshwater input to the surrounding ocean. Here we show, using a 3D ice shelf-ocean boundary current model, that deeper basal channels can lead to a significant amplification in channelized basal melting, meltwater channeling, and warming and salinization of the channel flow. All of these channelized quantities are also modulated by channel width, with the level of modulation determined by channel height. The explicit quantification of channelized basal melting and the meltwater transport in terms of channel cross-sectional shape is potentially beneficial for the evaluation of ice shelf mass balance and meltwater contribution to the nearshore oceanography. Complicated topographically controlled circulations are revealed to be responsible for the unique thermohaline structure inside deep channels. Our study emphasizes the need for improvement in observations of evolving basal channels and the hydrography inside them, as well as adjacent to the ice front where channelized meltwater emerges.

Ice shelves, which fringe the Antarctic ice sheet and are fed by grounded ice from upstream, buttress ice outflow, limiting the ice discharge from the inland ice sheet to the ocean[1,2]. The increasing intrusion of warm ocean waters onto the West Antarctic continental shelf has been linked to accelerated ice-shelf basal melting[3,4]. The enhanced basal melting causes, in turn, thinning of the ice shelves, reduction of buttressing, and acceleration of ice flow, and ultimately leads to an increased Antarctic contribution to sea level rise[5,6]. Basal channels that are variously ocean-sourced, subglacially sourced, and grounding line-sourced are widespread on Antarctic ice shelves[7,8]. The formation of ocean-sourced channels is controlled by oceanic processes within the sub-ice cavities while grounding line-sourced channels are initiated by lateral variations in topography at the grounding line and enhanced by ocean-driven melting. Subglacially sourced channels are, in contrast, formed by subglacial freshwater discharge at the grounding line. Channels directly influence ice-shelf stability by interacting with ice-shelf fractures,

leading to along-channel retreat[8,9] and frontal calving[10,11]. The existence of channels can also control ice-shelf mass balance by redistributing the basal melting across the ice shelf[12–14]. In addition, basal channels frequently end in persistent open-water polynyas at the ice front[8,15–17], suggesting that channelized outflow can have enough buoyancy and heat energy to reach the surface with a temperature that remains above the surface freezing point, despite the change in pressure and vigorous mixing with cold surface waters[18,19].

Channelized outflows are sourced from a combination of glacially modified water (GMW)[8,20], a mixture of warm ocean waters and resulting meltwater, and any addition of subglacial freshwater from the grounding line[21,22]. The channelized flow supports basal melting, enhancing the channel structure[23], and promotes basal and surface fractures by altering the stress distribution[24,25], which may reduce the structural integrity of the ice shelves. The basal channels, in turn, regulate the dynamics and properties of the subice circulation and,

[1]Southern Marine Science and Engineering Guangdong Laboratory (Zhuhai), Zhuhai 519080, China. [2]Department of Geography and Environmental Sciences, Faculty of Engineering and Environment, Northumbria University, Newcastle upon Tyne NE1 8ST, UK. [3]British Antarctic Survey, Cambridge CB3 0ET, UK. [4]School of Marine Sciences, Nanjing University of Information Science and Technology, Nanjing 210044, China. [5]These authors contributed equally: Adrian Jenkins, Paul R. Holland. ✉ e-mail: chengchen@sml-zhuhai.cn

thus, the patterns of ice-shelf basal melt[12–14,26]. As a consequence, increased ice shelf basal roughness via topographic features that span a range of length scales strongly correlates with increased basal melt[27]. Constraining the patterns and rates of basal melt associated with basal channels has been suggested as a key research priority, which depends critically on understanding the process-based controls on the dynamics and thermohaline structure of channelized GMW flows[7].

Depth-integrated plume models have been an effective numerical tool to investigate the mutual dependence of buoyancy-driven GMW flow and basal channel evolution[12–14]. However, growing evidence from recent high-resolution simulations[28–30] has demonstrated the importance of capturing the vertical shear and thermohaline structure of subice boundary currents in order to replicate heat and salt transport from the ambient ocean across the current to the ice shelf base. Specifically, the turbulent mixing of heat and salt across the ice–ocean boundary layer and across the pycnocline separating GMW and ambient ocean control the melting-induced cooling and freshening and the entrainment-induced warming and salinization, respectively, of the boundary flows. These processes depend intrinsically on resolving the details of the vertical stratification and shear[31,32]. This cannot be achieved in depth-integrated plume models, in which physical quantities are assumed to be vertically uniform across the plume and discontinuous at its edges (ref. 33, and derivatives thereof). To overcome this limitation, regional sub-ice cavity models[34–36] have been coupled with ice sheet models. However, these ocean circulation models currently lack the necessary vertical resolution to capture the structure of ice shelf−ocean boundary currents (ISOBCs), i.e., the upper relatively well-mixed boundary layer and the underlying pycnocline where properties transition to those of the ambient water, while even coarser resolution[37] or parameterization of the entire subice circulation[38] are typically employed in circum-Antarctic modeling. While DNS[39] (direct-numerical-simulation) and LES (large-Eddy-simulation)[28,29,40–42] ice–ocean boundary layer models may be potential candidates in the future, which can adequately resolve the ISOBC structure and have been used for process studies, they have thus far been unable to account for complex, large-scale basal topographic features owing to the formidable computational cost associated with their much higher resolution (in both vertical and horizontal directions).

Consequently, there is a pressing need to develop 3D ISOBC models. This can be a useful compromise with an adequate vertical resolution to capture the key processes occurring at both the upper (ice–ocean) and lower (entrainment) boundaries of the channelized currents, but computationally efficient enough to be coupled with existing ice sheet models to investigate the interactions between oceanic processes and basal channel evolution. Such an ISOBC model would represent a further development of the current hierarchy of models from vertical 1D[43–45] to 2.5D (with across-slope gradients of all variables omitted) vertical slice models[46] that have been used to investigate the fundamental dynamic and thermodynamic vertical structures of the ISOBC. Those models suggest an upper, relatively well-mixed, turbulent boundary layer (the plume region), a broad pycnocline characterized by weaker mixing, and exterior geostrophic flow. However, these low-dimensional ISOBC models cannot simulate the full spatiotemporal influence of ice shelf basal channels on the dynamics and thermohaline structure of the ISOBC.

In this study, we investigate the interactions between the ISOBC and an ice shelf basal channel, as represented in a 3D ISOBC model. The impacts of individual basal channels with varied cross-sectional shapes (CSS) on channelized melting and GMW channeling are systematically studied with the model.

## Results
### Experimental design
Previous studies have shown the importance of basal channels beneath the Pine Island ice shelf in distributing basal melting[47] and in steering

and then channeling meltwater out to open waters[15–17]. We therefore apply our 3D ISOBC model to investigate channel-influenced currents and basal melting underneath a Pine-Island-like ice shelf by conducting a suite of idealized numerical experiments (Methods). While Pine Island sets the overall scale of the domain, we nevertheless model an individual basal channel in order to study fundamental behavior rather than to reproduce the details of the complicated sub-ice ocean circulation and associated melt rate distribution beneath the Pine Island ice shelf.

The model uses a uniform vertical resolution of 4 m, which is intermediate between present ice-cavity models and LES[40]. All the simulations are forced by a constant GMW buoyancy influx (2.3 m$^4$ s$^{-3}$; Methods) along a stretch of the deepest (southern) boundary, although the basal melting over the whole domain soon becomes the predominant buoyancy source. In our reference run, for instance, the outflowing channelized buoyancy discharge (Methods) overtakes that initial GMW buoyancy influx within 14 h of the first arrival of the channelized GMW at the northern boundary and subsequently reaches a quasi-steady value of 296 m$^4$ s$^{-3}$. The ice shelf base is set to be planar with a prominent basal channel having a constant sinusoidal CSS in the along-slope direction (see, e.g., Fig. 1c, d).

Our idealized numerical experiments consist of one reference run and 20 sensitivity runs in total (Methods). The reference run provides the basic basal melting and upper plume patterns and channelized ISOBC structure, while the sensitivity runs, divided into S1–3 classes, examine the sensitivity of the reference results to a variety of factors, including channel CSS, ambient properties, overall basal slope, and vertical resolution (detailed in Table 1). In S1, various channel CSSs with different aspect ratios (i.e., the ratio of channel height to width) are used to investigate the role of channel geometry in controlling the distribution of melting and the overall GMW channeling capacity. These two quantities are crucial for ice shelf stability and GMW input into the surrounding ocean. These basal channel geometries are characteristic of ocean- or grounding line-sourced channels and have kilometer-scale widths. Sub-kilometer-wide basal channels cannot be considered in this study owing to the limits of the horizontal resolution limitation (Methods). Using a horizontally high-resolution 3D ice-cavity model, Millgate et al.[26] found that for narrower channels, the geostrophic circulation established within wider channels transitions to a slower, ageostrophic overturning circulation. The simulated basal melt rate is then no longer sensitive to the aspect ratio and is quite limited. In further sensitivity experiment classes, we examine the impact of changing ambient properties (S2-A), overall basal slope (S2-S), and vertical resolution (S3) on channelized melting and channeling capacity.

All simulations cover 1800 h, sufficient to achieve a quasi-steady state (QSS). The simulated results from 1080 to 1800 h (i.e., 1 month), representative of that QSS (shown later), are used in the subsequent analyses. Accordingly, a time-accumulated basal melt is derived by integration from 1080 to 1800 h and then converted to a melt rate in meters per year through multiplication by 12. Similarly, the term "time-averaged" henceforth corresponds to averaging over this 1-month period.

### Basic patterns and channelized ISOBC structure
Here we refer to the H140W8 simulation (Table 1) as the reference run, while a no-channel test (H0) is carried out for comparison. In H0, after the GMW inflow is initiated, the buoyant ISOBC flows upslope and to the west due to the Earth's rotation (we select $f < 0$ as most ice shelves are in the Southern Hemisphere). After this initial phase, the ISOBC is gradually activated over the entire domain through ice-ocean interactions in initially quiescent regions and evolves to be well-developed in the last 30 days. That is the general behavior for all simulations, and we focus only on the QSS rather than the transitional period before that. The ISOBC becomes geostrophically adjusted so that it largely follows the straight isobaths in H0 (Fig. 1b). Most basal melt occurs

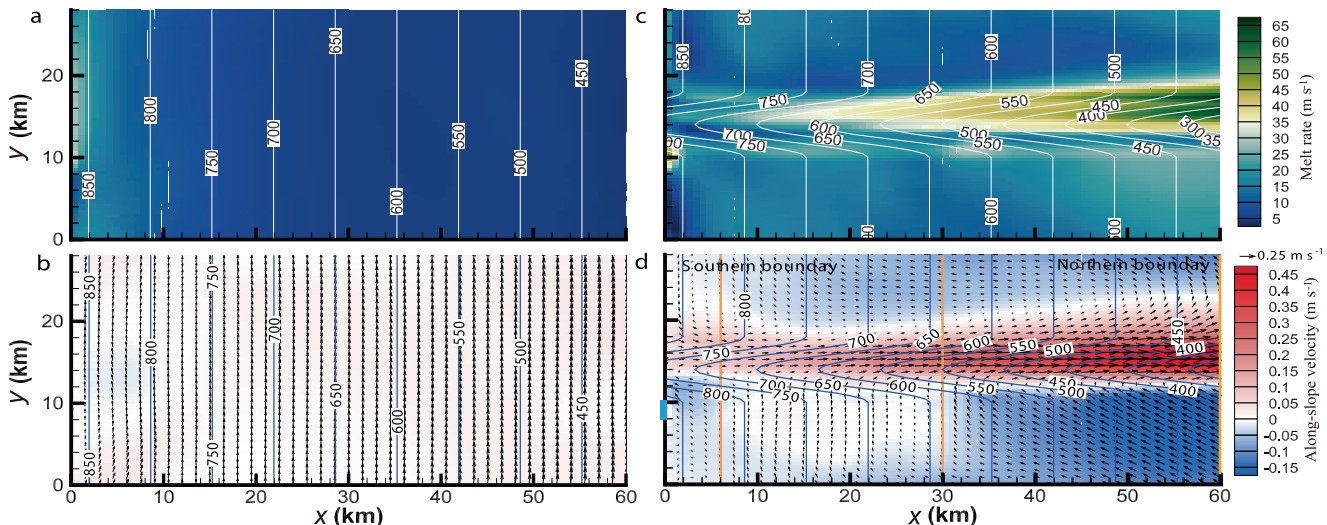

Fig. 1 | Comparison of basal melting and circulation patterns in the cases with and without a basal channel. Distribution of a, c basal melt rate and b, d time-averaged plume (see Methods for its thickness definition) circulation (black arrows; along-slope velocity is also shown by shading) in a, b H0 and c, d H140W8. The white and blue contour lines in (a, c) and (b, d), respectively, indicate the ice shelf draft. In d, the sky blue block at the southern boundary from y = 8 to 10 km indicates the inflow zone. The orange lines denote the across-slope sections shown in Fig. 2. A reference vector is shown immediately above the contour legend of along-slope velocity.

deeper than 800 m and is prominent to the west of the inflow along the southern boundary (Fig. 1a).

In contrast, as shown in Fig. 1d, the basal channel introduced in H140W8 redirects the ISOBC and channels it all the way to the northern boundary of the domain. The ISOBC within the channel undergoes

geostrophic adjustment, where the flow is governed by a balance between Coriolis force and the pressure gradient caused by isopycnals that are shaped by the topography of the basal channel. From the deepest part of the ice shelf, the ISOBC accelerates downstream and becomes focused along the western side of the channel. Towards the ice front (i.e., the northern boundary), the GMW flow exceeds the channeling capacity, and a flow out of the channel across its western boundary develops. Apart from a small patch to the west of the inflow, the basal melting occurs overwhelmingly over the western side of the channel shallower than about 650 m depth (Fig. 1c), consistent with the channeling of the ISOBC shown in Fig. 1d and visibly larger than the basal melt in H0. Concentrated channelized basal melt has been identified in high-resolution satellite and airborne observations[20,23] and modeling efforts[12,14,26]. More complex channelized melting patterns have also been found in field surverys[48,49], associated with more complex variations in the ice draft than assumed here[7,12,14,26].

Three East–West across-slope sections at x = 6, 30, and 60 km (orange lines in Fig. 1d) are selected to provide more information on the spatial evolution of the ISOBC structure along the basal channel (Fig. 2). The ISOBC core gradually develops along the channel, leaning on the western flank. As the buoyancy increases, the tilt of the isopycnals becomes more pronounced (Fig. 2a, d, g), leading to a higher velocity (Fig. 2b, e, h) through the thermal wind balance. With the accumulation of meltwater from upstream (Fig. 1d), the across-slope GMW flow eventually fills the channel (Fig. 2c) and crosses it once the channel capacity is exceeded (Fig. 2f, i). Meanwhile, a weak eastward (negative) across-slope velocity (indicated by the blue arrow in Fig. 2i) underneath the western channel flank indicates a topographic recirculation, a feature discussed in detail in the next section. The deviation of the vertical structure of the ISOBC outside the channel from that presented by Jenkins[43] may be attributed to the nonuniformity of the lateral flow field shown in Fig. 1d. These results show that the 3D ISOBC model presented here can provide new insight into the vertical dynamic and thermodynamic structures of a channelized ISOBC.

## Importance of basal channel cross-sectional shape in ice-ocean interactions

The purpose of the S1 experiments is to systematically investigate and quantify the influence of basal channel CSS on the channelized basal

## Table 1 | Explanatory notes on the design of experiments

| Class | Nomenclature | Description |
|---|---|---|
| Ref. | H140W8 | The basal channel is 140 m in height and 8 km in width; ambient salinity and temperature are 34.36 psu and 0.18 °C, respectively; the ice-shelf basal slope is 0.0075 |
| S1 | H0 | Same as H140W8 but with no basal channel |
| | H60W4 | Basal channel is 60 m in height and 4 km in width |
| | H60W8 | Basal channel is 60 m in height and 8 km in width |
| | H60W12 | Basal channel is 60 m in height and 12 km in width |
| | H140W4 | Basal channel is 140 m in height and 4 km in width |
| | H140W12 | Basal channel is 140 m in height and 12 km in width |
| | H220W4 | Basal channel is 220 m in height and 4 km in width |
| | H220W8 | Basal channel is 220 m in height and 8 km in width |
| | H220W12 | Basal channel is 220 m in height and 12 km in width |
| | H300W4 | Basal channel is 300 m in height and 4 km in width |
| | H300W8 | Basal channel is 300 m in height and 8 km in width |
| | H300W12 | Basal channel is 300 m in height and 12 km in width |
| S2-A | Sa34.31 | Ambient salinity is decreased to 34.31 psu; the corresponding temperature is set to 0.05 °C according to Gade line |
| | Sa34.41 | Ambient salinity is increased to 34.41 psu; the corresponding temperature is set to 0.33 °C according to Gade line |
| S2-S | Sp05 | Ice-shelf basal slope is decreased to 0.005 |
| | Sp11 | Ice-shelf basal slope is steepened to 0.011 |
| S3 | H60W8-C | Same as their counterparts in S1, but vertical spacing is coarsened from 4 to 10 m |
| | H140W8-C | |
| | H220W8-C | |
| | H300W8-C | |

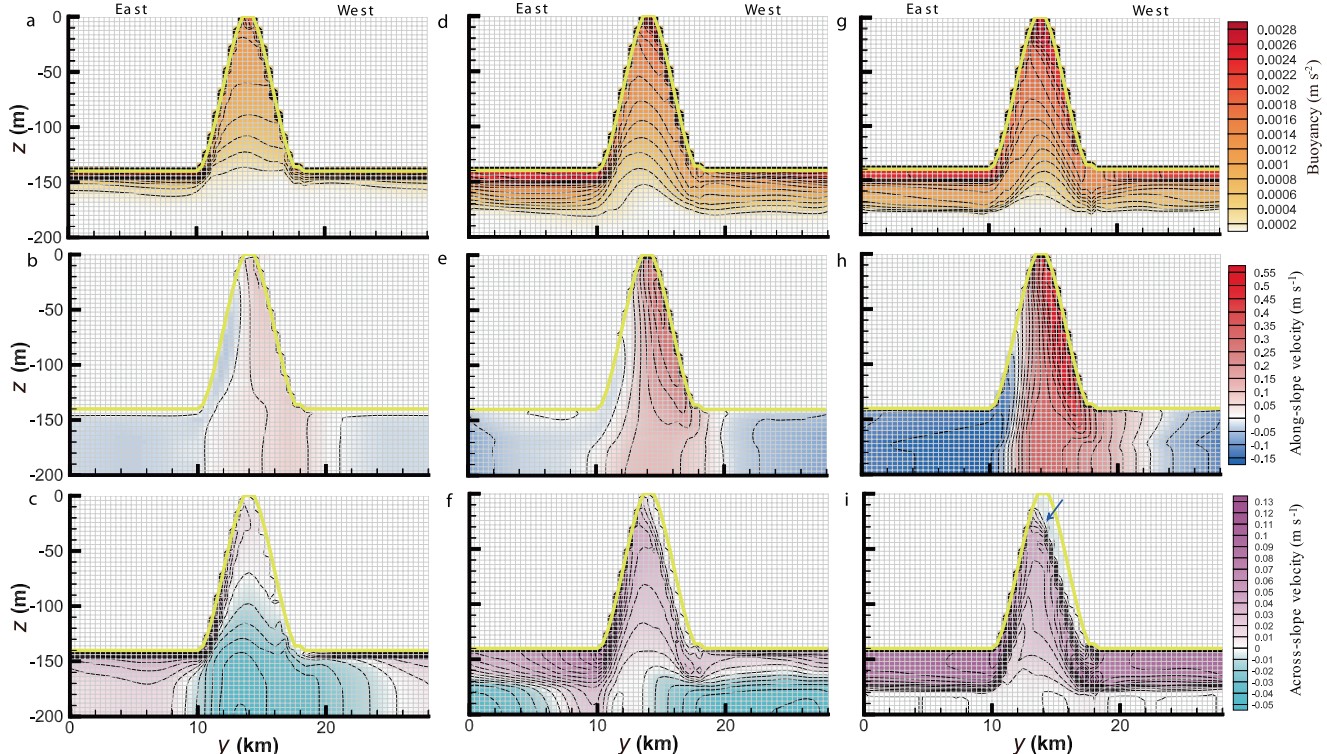

**Fig. 2 | Evolution of the vertical structure of the ice shelf–ocean boundary current (ISOBC) within a channel.** Across-slope distribution of time-averaged **a**, **d**, **g** buoyancy (i.e., the reduced gravity as defined in Methods), **b**, **e**, **h** along-slope velocity, and **c**, **f**, **i** across-slope velocity in H140W8 at $x$ = **a**–**c** 6, **d**–**f** 30, and **g**–**i** 60 km (orange lines in Fig. 1d). In **i**, the blue arrow indicates the presence of a topographic recirculation adjacent to the western flank of channel. The ice base in each plot is marked by the yellow line, and the model grid by the mesh. The staircase-like patterns adjacent to the ice base are artifacts of the structured $z$ grid in our ISOBC model.

melt and outflowing heat and buoyancy discharge. The melt rates, area-averaged over the channel domain, exhibit a strong dependence on channel CSS, characterized by changes in channel width, height, and aspect ratio. For the shallowest (60 m) cases, the channelized basal melt is insensitive to channel width and is weaker than for the deeper cases (Fig. 3a). For the intermediate height (140 and 220 m) cases, the channelized basal melt increases with channel width but plateaus when the channel widens beyond 8 km. For the deepest (300 m) cases, the channelized basal melt increases monotonically with channel width. As shown in Fig. 3b, the channelized melt increases more rapidly with channel height as the channel widens, but again the plateauing effect mentioned above means that the differences between the 8 and 12 km wide channels are small (Fig. 3a). Considering instead, the relationship between melt rate and channel aspect ratio (Fig. 3c), we find reasonably linear relationships for all widths, with the slope of the regression line increasing with channel width (Fig. 3d), while the intercept is similar in each case (Fig. 3c), but well above the 7.9 m y$^{-1}$ found for the ice shelf base with no channel in H0 (Fig. 3b, c). Combining the relationships deduced from Fig. 3c, d, an empirical function describing the dependence of the channelized basal melt rate on CSS can be derived, as shown in Fig. 3e. The value of the universal intercept, 20.16 m y$^{-1}$, is determined by best matching the simulated melt. This quantification demonstrates that both channel height and aspect ratio regulate the overall magnitude of channelized basal melt, although the applicability of the relationship needs to be tested for a wider range of CSS. An obvious manifestation of the limits of the relationship is the discrepancy between the derived intercept and the melt rate derived from the H0 simulation. Clearly, the linearity of the derived relationship breaks down for very narrow channels, presumably as a result of the transition in circulation mode to the ageostrophic overturning discussed by

Millgate et al.[26] that cannot be simulated here because of the limitation of horizontal resolution.

We may examine why channelized basal melt responds to variations in channel CSS by considering the determinants of basal melting, that is, the near-ice temperature and friction velocity. H60W12, H300W12, and H300W4 are selected, illustrating the largest differences in melt caused by changes in the channel height and width (Fig. 3c). Considering the height-induced difference, the channelized melt is enhanced markedly in extent and magnitude in H300W12 (Fig. 4d), compared with that in H60W12 (Fig. 4a), consistent with the greater friction velocity and higher near-ice temperature in H300W12 (Fig. 4e, f), compared with that in H60W12 (Fig. 4b, c). Considering width-induced differences, the extent of the main channelized melt is greatly reduced when the channel narrows from 12 to 4 km (Fig. 4g), and there is an associated reduction in friction velocity (Fig. 4i). A steeper eastern channel flank induces stronger mixing of GMW with ambient water, resulting in reduced buoyancy and hence lower speed of the main channel flow. Nevertheless, the near-ice warming (Fig. 4h) caused by the deepening basal channel maintains higher melt (Fig. 4g) than in H60W12 (Fig. 4a). This suggests that channel height determines the melt magnitude by controlling both friction velocity and near-ice temperature, while channel width influences the melt extent by constraining the ISOBC dynamics. Because the sidewall slope of channels depends on both channel height and width, the channelized melt rate, representative of the robustness of main channel flow, becomes insensitive to wider channels (8 and 12 km) for runs with channel shallower than 300 m (Fig. 3a).

In addition to channelized basal melting, we quantify the magnitude and properties of the channelized outflows that are of particular concern for assessing the influence of ice shelf basal meltwater on near-coastal waters[50,51], the broader Southern Ocean[52,53], and the global

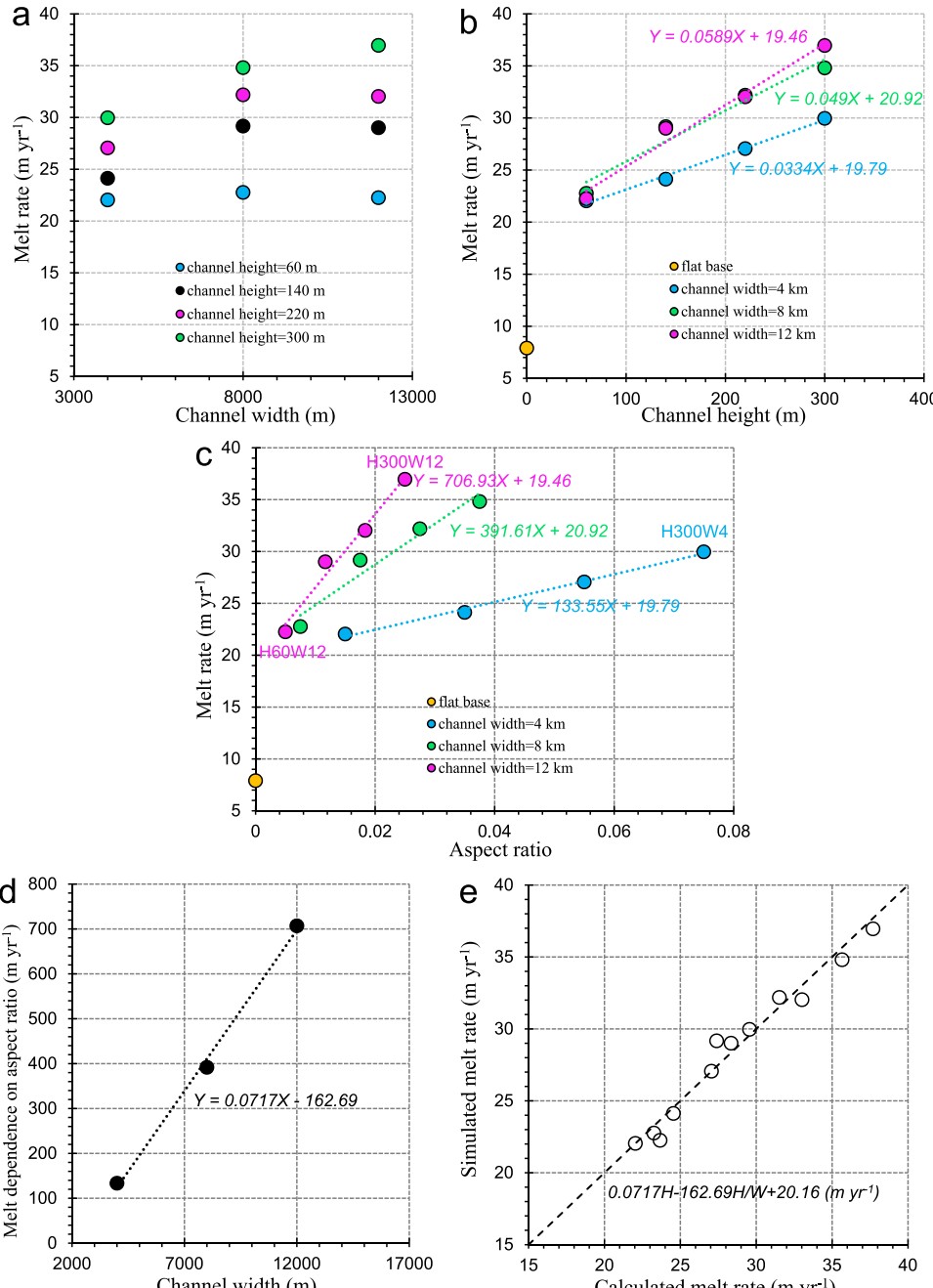

**Fig. 3 | Channelized basal melting and its quantitative dependence on channel cross-sectional shape (CSS).** Relationship of channelized basal melt rate with channel **a** width, **b** height, and **c** aspect ratio with respective linear regressions. The melt rate for the flat case (H0) also shown in (**b**, **c**). **d** Relationship between the slope derived from linear regression shown in (**c**) and channel width. **e** Comparison between calculated channelized basal melt rates using the indicated empirical function of channel CSS and simulated values.

climate system[54–56]. The discharge-averaged heat of the outflow (see Methods) for all S1 runs exhibits the quasi-steady value, which is dependent on the channel CSS (Fig. 5a). The heat is most sensitive to channel height, rising as the height increases, although the sensitivity also diminishes as the channels deepen. In contrast, the influence of channel width is more limited, although largest for the deepest channel, ranging from +3.6% for the 4-km wide channel to −4.7% for the 12-km wide channel relative to the 8-km wide channel.

Looking at the vertical thermal structure in the across-channel section at the northern boundary for differing channel height but fixed 8 km width (henceforth referred to as intermediate-width runs), we find that a fully developed relatively cold (lower than −1.4 °C) and well-

mixed layer builds up underneath the ice base across the full width of the shallowest channel (Fig. 5b). The upper interface of the underlying thermocline is curved up into the channel, following the channel geometry, but the lower interface is, in contrast, both flatter and vertically sharper. When the basal channel deepens to 140 m, the upper mixed layer becomes warmer and discontinuous across the channel (Fig. 5c). The upper interface of the thermocline intersects the ice base immediately east of the channel apex, and its lower interface has a weaker stratification and intrudes upwards into the channel. When the channel further deepens (Fig. 5d, e), the meltwater plume along the eastern flank becomes even warmer, while the cold layer on the western flank warms and thins, becoming increasingly confined to the

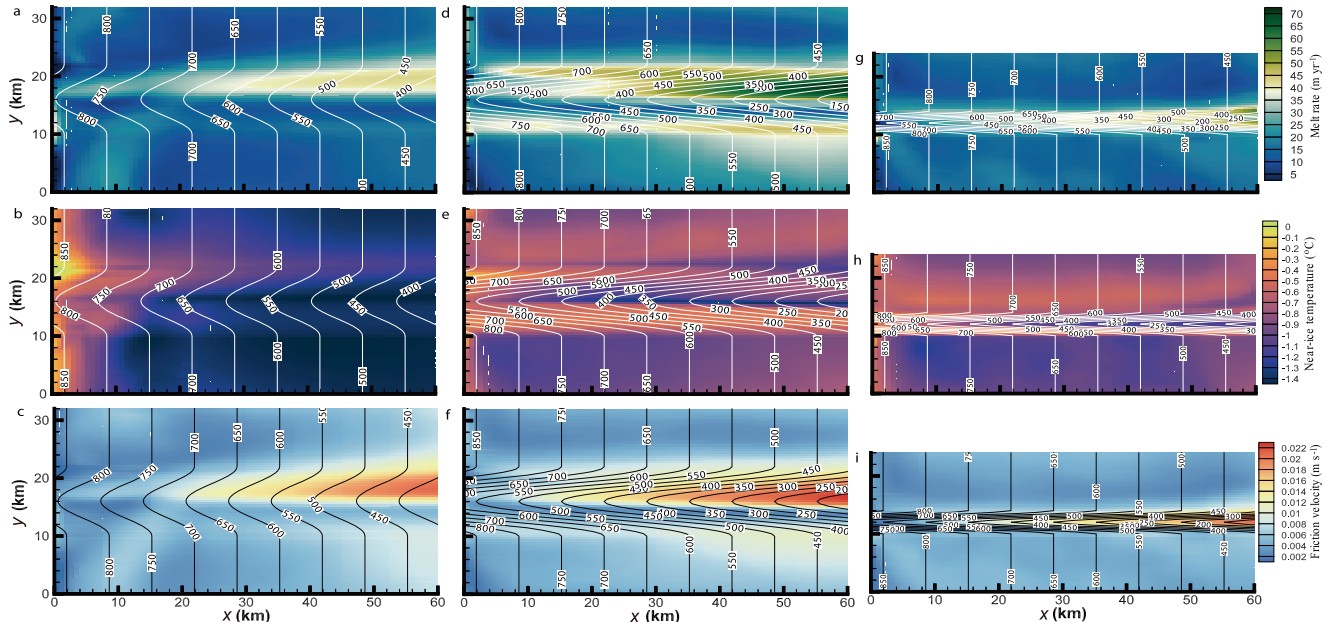

**Fig. 4 | Differential basal melt and its determinants caused by differential channel cross-sectional shape.** Distribution of **a, d, g** basal melt rates, time-averaged **b, e, h** near-ice temperature and **c, f, i** friction velocity in **a**–**c** H60W12, **d**–**f** H300W12, and **g**–**i** H300W4. The white and black contour lines in (**a, d, g, b, e, h** and **c, f,** i), respectively, indicate the ice shelf draft. Note that the extent of the planar base part adjacent to each side of the channel areas is consistently set to 10 km for all runs here.

apex of the channel with the underlying thermocline sharpening to the east of the apex. These changes in the vertical thermal structure are consistent with those in the vertical haline (Supplementary Fig. 1) and thus stratification structures (Supplementary Fig. 2a–d). The density of cold Antarctic waters is conventionally described by a linearized equation of state in which the haline contraction effect dominates over thermal expansion[33,43–46]. As a result, higher salinity typically corresponds to denser waters, and thus reduced buoyancy, and vice versa.

All these thermohaline changes are closely tied to the hydrodynamic changes within the channels as they deepen. Two key processes contributing to vertical heat transport, vertical diffusion and vertical advection (Fig. 6), are considered. The distribution of parameterized vertical diffusivity and resolved vertical velocity in two across-channel sections at $x = 30$ and 60 km within the well-developed regime are averaged for the following analysis. When the basal channel is shallow (i.e., in H60W8), the vertical diffusivity is high everywhere beneath the ice (Fig. 6a), consistent with the well-defined upper mixed layer that forms (Supplementary Fig. 2a). The vertical velocity displays a positive-to-negative symmetrical pattern within the mixed layer, indicative of a flow passing across the shallow channel (Fig. 6e).

When the channel is deepened to 140 m (i.e., in H140W8), the lengthening and steepening of the eastern channel flank leads to enhanced diffusivity (Fig. 6b), where the rapid inflow of GMW shears past the ambient water (Supplementary Fig. 2b). That mixing substantially warms (Fig. 5c) and salinizes (Supplementary Fig. 1b) the GMW, so the resultant water mass is referred to as intermediate GMW (iGMW) here and generates the pycnocline structure discussed above (Supplementary Fig. 2b). Associated with the geostrophic along-slope velocity on the western side of the channel (Fig. 2), an eastward Ekman transport develop adjacent to the western channel flank (Figs. 2i and 6f). The emergence of this feature can be regarded as a dynamic response to a larger across-channel pressure gradient induced by the more tilted isopycnals and steeper topography (Supplementary Fig. 2b) and the associated stronger along-channel geostrophic flow relative to the shallow (60 m) channel case (Supplementary Fig. 2a). While most of the iGMW can still cross the channel (Fig. 2f, i), some iGMW is, however, recirculated by the eastward Ekman transport,

which makes the water interacting with the western channel flank warmer (Fig. 5c) and saltier (Supplementary Fig. 1b) than that in H60W8 (Fig. 5b and Supplementary Fig. 1a). This recirculation, combined with the faster main channel flow, leads to higher melting of the western channel flank. The resultant strong stratification (Supplementary Fig. 2b) suppresses local turbulent diffusivity (Fig. 6b).

When the channel is deepened further (i.e., in H220W8 and H300W8), the processes discussed are further enhanced, leading to rapid mixing between GMW and ambient water along the eastern channel flank (Fig. 6c, d) and the recirculation of even warmer (Fig. 5d, e) and saltier (Supplementary Fig. 1c, d) iGMW upward along the western channel flank (Fig. 6g, h), resulting in the low vertical diffusivity there (Fig. 6c, d). The progressive recession of lighter GMW and the advance of denser iGMW conspire to form the pinched isopycnals (Supplementary Fig. 2c, d) noted within the deepest channel. These thermodynamic processes are responsible for the described changes in the thermohaline structure as the basal channels deepen.

The discharge-averaged heat of the channelized outflow has been shown to depend largely on basal channel height rather than channel width (Fig. 5a). The dependence on height for the intermediate-width runs can be described by a log law, allowing for both a concise form and its more desirable performance than a power law (blue component in Fig. 7a), which would allow the quantification of the heat discharge of the channelized outflows (Methods), if we could formulate a relationship between the outflowing discharge (Methods) and the channel CSS. That latter step is challenging, although the channelized outflowing discharge p.u.w. (per unit width) shows a clear correlation with channel CSS (Fig. 7b), increasing with channel height and becoming more sensitive to channel width as height increases. Using the intermediate-width runs as a reference, the normalized discharge p.u.w. is quantified reasonably as a function of channel height via a power-law relationship with an exponent (0.79) smaller than one (red component in Fig. 7a). As shown in Fig. 7c, even though the dependence of the normalized discharge p.u.w. on channel aspect ratio is sensitive to channel width, there still exists an overall linear relationship between them. Combining the three functions presented in Fig. 7a, c, the channelized discharge p.u.w. and heat discharge p.u.w.

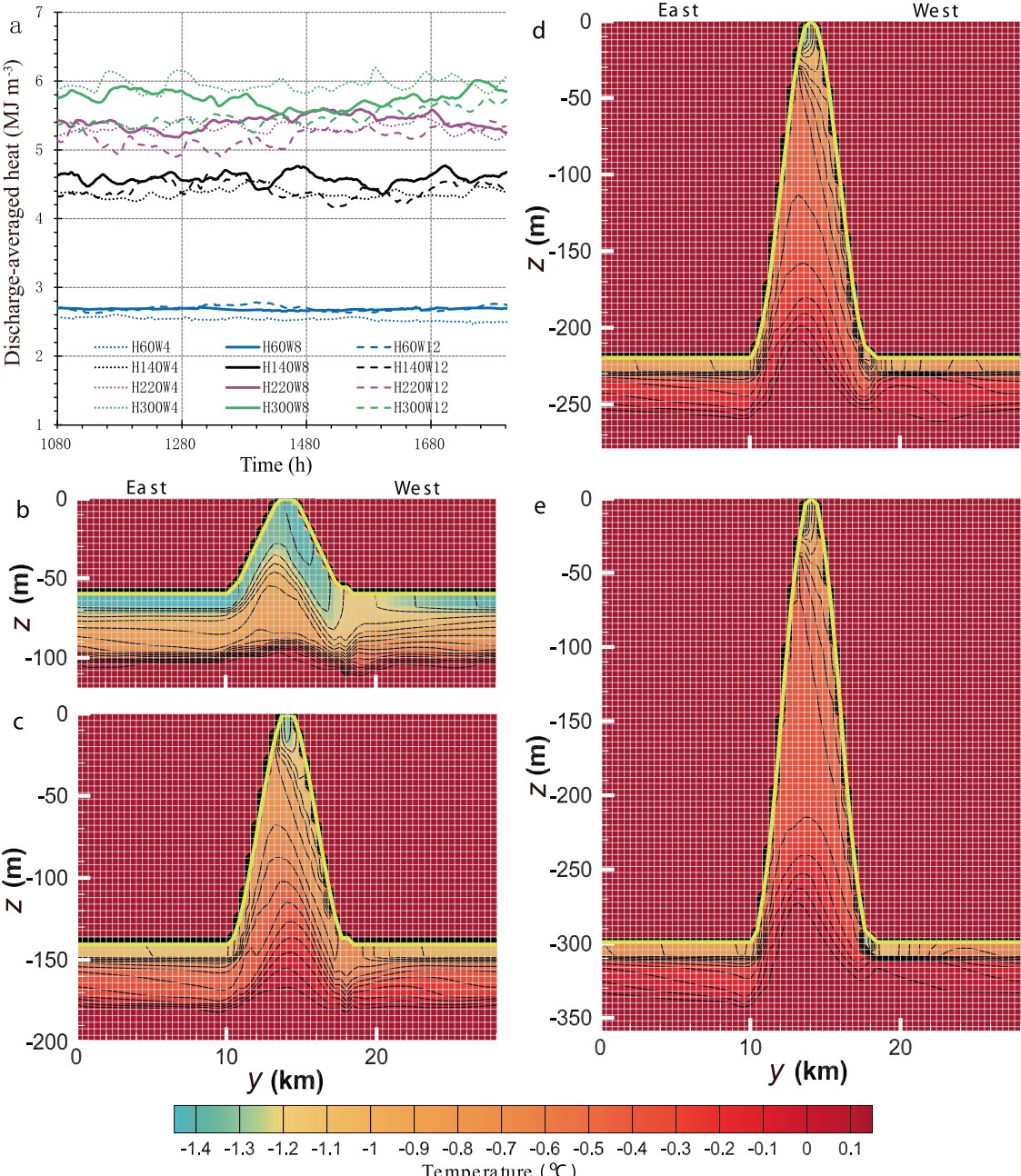

**Fig. 5 | Channelized outflowing heat. a** Discharge-averaged heat of channelized outflow for S1 runs, as well as H140W8, during the quasi-steady period. Time-averaged vertical thermal structure in the cross-section at the northern boundary in **b** H60W8, **c** H140W8, **d** H220W8, and **e** H300W8. The ice base in **b**–**e** is marked by the yellow lines, and the planar parts of the ice base correspond to the same ice draft. The default value of 0.18 °C inside the solid ice (i.e., above the planar ice base and outside the channel region) has no physical significance. Model grids are also indicated by the mesh.

can be quantified in terms of channel height and aspect ratio, indicating that deep basal channels with steep side slopes can lead to large mass and heat discharge p.u.w., compared with shallow channels with more gradual side slopes. Then by multiplying channel width by an empirical constant (0.833), determined by best matching the simulated heat discharge (Fig. 7d), an expression for heat discharge as a function of channel CSS is derived, as labeled in Fig. 7d. An expression for quasi-steady channelized discharge is also given in Fig. 7c. Quantification of the buoyancy discharge of the outflow (Methods) is then possible, because the buoyancy and heat are linearly related in the case of mixing with a single source water mass[57] (see Fig. 7e). The resulting expression for the buoyancy discharge as a function of channel CSS is shown in Fig. 7f, using an empirical constant of 1.031 to provide the

best fit to modeled values. Finally, we conducted six additional model runs using higher resolution ($dy = 250$ m and/or $dz = 2$ m) to examine the reliability of the above relationships for the shallowest (60 m) and the narrowest (4 km) cases (see Supplementary Note 1). It is found that H60W4 has relatively the largest deviation among all six runs because of the issues discussed earlier in representing the smallest cross-sectional areas. To this end, an imperative effort needs to be made to extend the applicability of all these empirical expressions of channelized quantities beyond the present range of CSS.

## Sensitivity to ambient ocean properties

The purpose of the S2-A experiments is to quantify the influence of ambient water properties on the channelized melt and transport in view

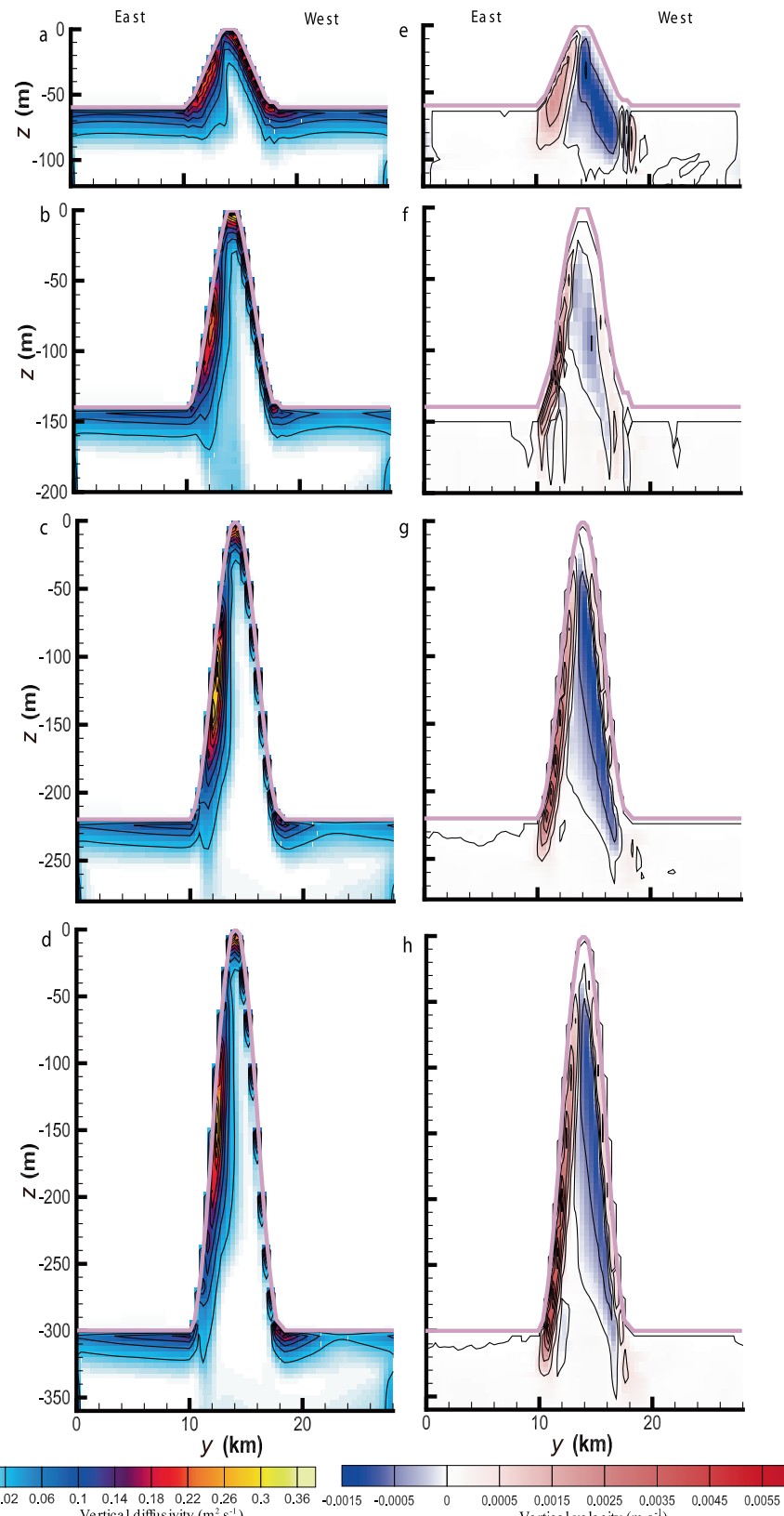

**Fig. 6 | Variables associated with channelized vertical heat transport for S1 runs.** Section-averaged (i.e., $x = 30$ and $60$ km) across-slope distribution of time-averaged **a**–**d** vertical diffusivity and **e**–**h** vertical velocity in **a**, **e** H60W8, **b**, **f** H140W8, **c**, **g** H220W8, and **d**, **h** H300W8. The ice base in each plot is marked by the pink line, and the planar part of the ice base corresponds to the same ice draft.

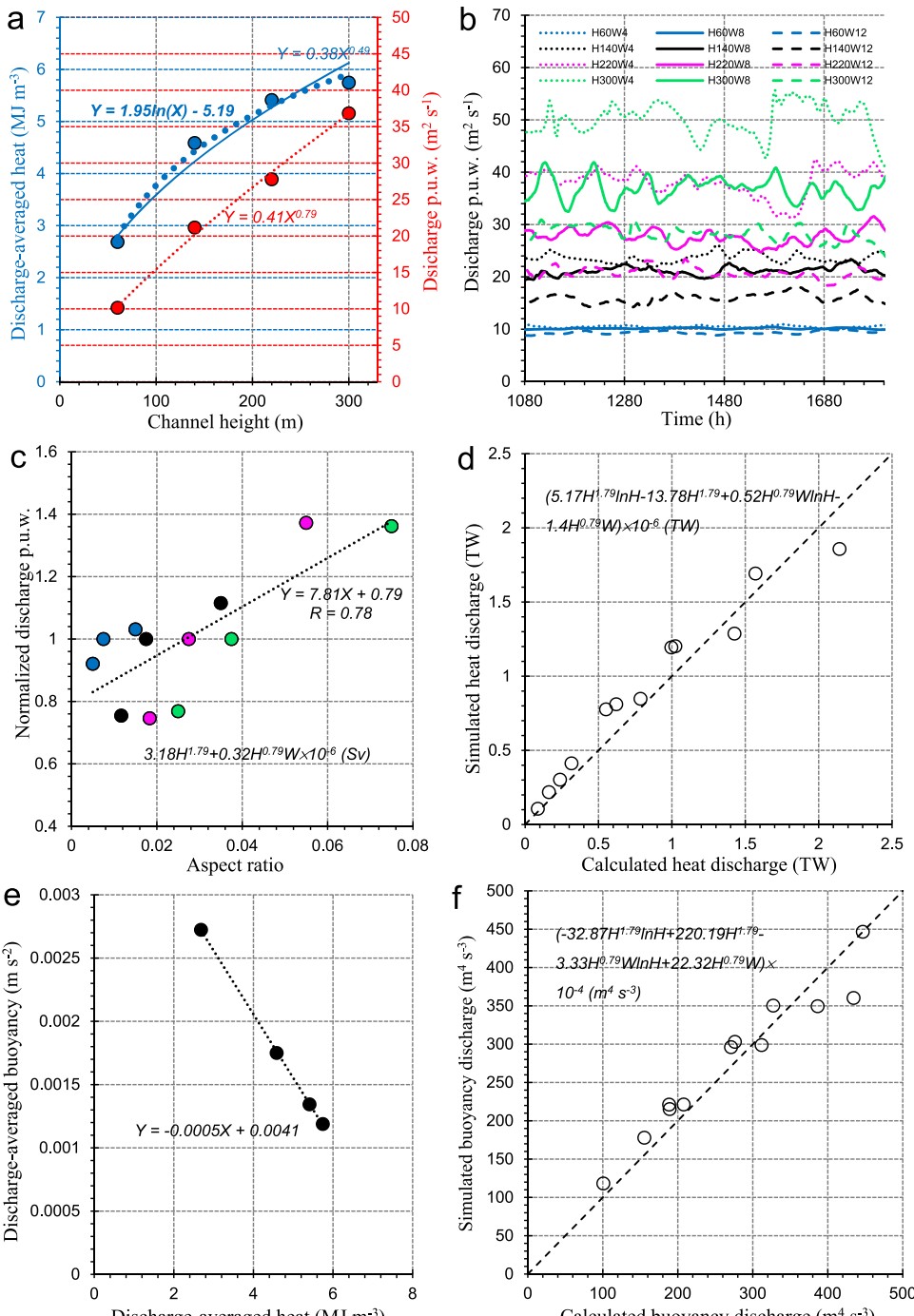

**Fig. 7 | Quantification of the relationship between heat discharge of channelized outflow and channel cross-sectional shape (CSS). a** Nonlinear relationship of time- and discharge-averaged heat and discharge p.u.w. of channelized outflow against channel height for runs with 8 km width in S1. **b** Time series of discharge p.u.w. of channelized outflow in the quasi-steady state for S1 runs. **c** Linear relationship between discharge p.u.w. normalized using the corresponding values for 8 km width runs (red dots in (**a**)) and channel aspect ratio; the expression for channelized discharge is also provided. Comparison between calculated **d** heat and **f** buoyancy discharge of channelized outflow using the functions of channel CSS presented here and the corresponding modeled values. **e** Linear relationship between discharge-averaged buoyancy and heat.

of the crucial role of entrainment of ambient water[30,44,46]. Note that when all the ISOBC model runs to reach their QSS, the predominant buoyancy source, as mentioned before, comes from ice-shelf basal melting, with the contribution from the inflow buoyancy transport of little consequence. Warmer ambient water (runs Sa34.31, H140W8, and Sa34.41) results in some amplification in both the magnitude and extent of channelized melt (Supplementary Fig. 3-R1; C2–4), an effect that can be attributed mainly to near-ice warming (Supplementary Fig. 3-R2;

C2–4), the change in friction velocity being imperceptible (Supplementary Fig. 3-R3; C2–4). Similar to channelized melt, the channelized outflowing heat discharge increases monotonically with increases in ambient temperature (Supplementary Fig. 4a), because ambient warming increases the discharge-averaged heat (Supplementary Fig. 4c) but contributes little to the channelized outflowing discharge itself (Supplementary Fig. 4b). The buoyancy discharge behaves analogously to the heat discharge (Supplementary Fig. 4d, e).

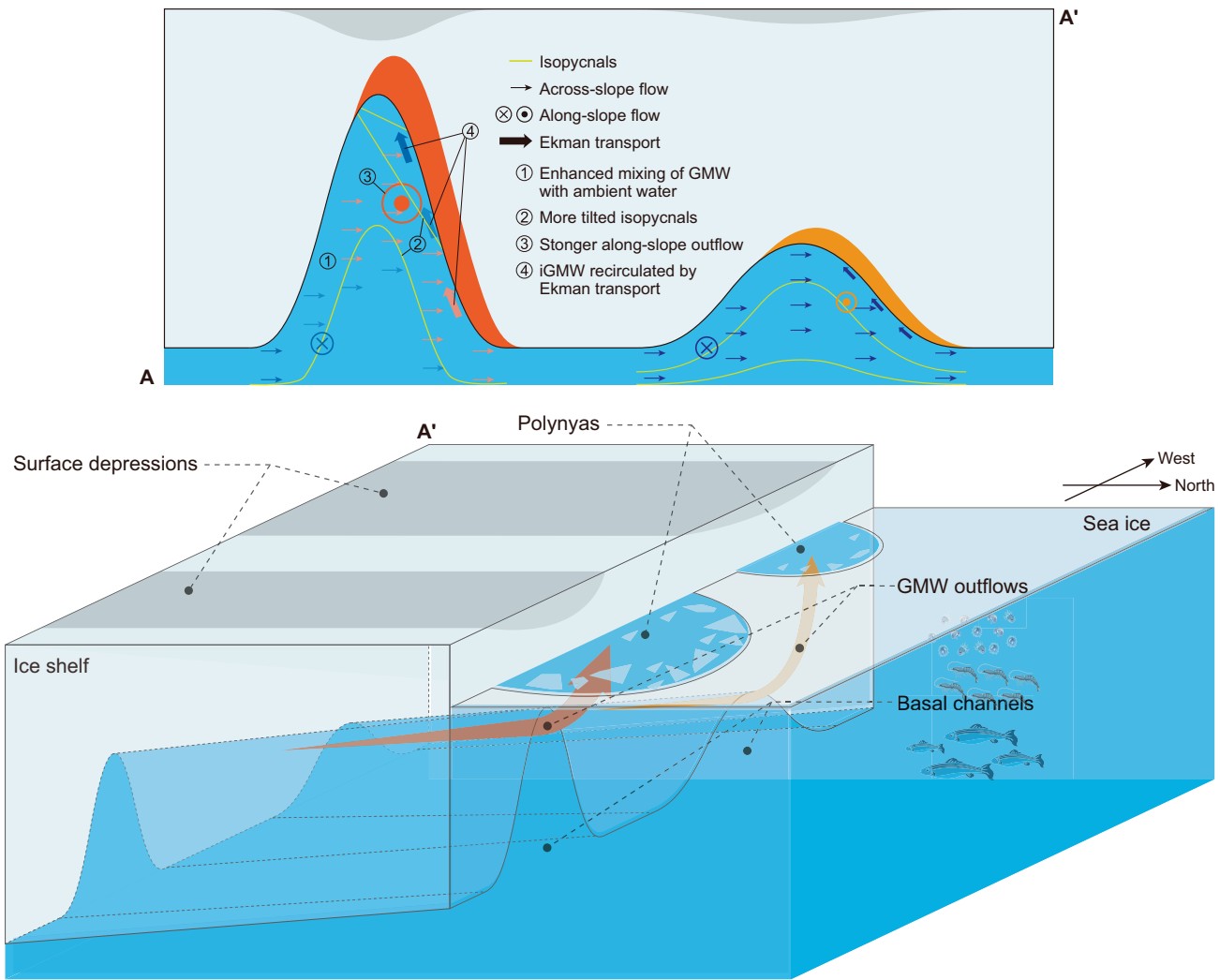

**Fig. 8 | Schematic representation of channelized ice shelf−ocean boundary currents.** The upper panel shows a comparison of critical physical processes underneath deep (left) and shallow (right) basal channels. Shallow channels have little influence on the properties of the across-channel glacially modified water (GMW) flow, with little GMW recirculated along the western flank of the channel. In contrast, the stratification and hydrodynamics are significantly changed in deep channels by four critical processes, as illustrated, leading to the enhancement of basal melt and meltwater channeling. The colored arrows transitioning from blue to pink denote the transformation of GMW to warmer and saltier intermediate GMW and vice versa. The lower panel illustrates the surfacing of warm channelized GMW, leading to the formation of coastal sensible-heat polynyas by melting fast ice adjacent to an ice shelf with a wealth of implications for the multidisciplinary Antarctic oceanography. Not to scale.

## Sensitivity to ice shelf basal slope

In addition to the channel CSS, the ice-shelf basal slope is another topographic determinant of buoyant meltwater plume forcing[43–46], so the sensitivity of channelized melt and transport to the basal slope is examined in S2-S. Steeper basal slope (runs Sp05, H140W8, and Sp11) leads to increases in both the magnitude and extent of channelized melt (Supplementary Fig. 3-R1; C1, 3, 5), driven by both near-ice warming (Supplementary Fig. 3-R2; C1, 3, 5) and friction velocity enhancement (Supplementary Fig. 3-R3; C1, 3, 5). Increasing basal slope also leads to an enhancement in the channelized outflowing heat (Supplementary Fig. 4a) and buoyancy discharge (Supplementary Fig. 4d), but the causality is opposite from that which controls the sensitivity to ambient properties; steeper basal slope augments the channelized discharge (Supplementary Fig. 4b) but contributes little to the discharge-averaged heat (Supplementary Fig. 4c) and buoyancy (Supplementary Fig. 4e). If variations in the channelized melt and transport caused by varying basal slope and ambient properties are assumed to be independent, the applicability of the relationships presented above could be extended by accounting for the linear changes apparent in these sensitivity studies (see Supplementary Fig. 5).

## Discussion

Channelized basal melting of ice shelves is shown here to be controlled by the channel CSS, which can be quantified in terms of channel height and aspect ratio (Fig. 3e). Remote sensing observations of basal channel geometry and channelized melting[8,20] could potentially be used to verify the empirical relationships presented here. Those relationships could then provide an alternative means (derived from the 3D ISOBC model) of inferring an overall channelized melt rate for the current Antarctic ice shelf basal topography.

Investigating the evolution of basal channel systems influenced by changing oceanic conditions requires the use of ice sheet-ocean coupled models rather than the stand-alone ice sheet models sometimes used (e.g., refs. 58,59). When stand-alone ice sheet models are used, an idealized, symmetric melt rate distribution across the channel is normally prescribed, typically using a Gaussian function centered at the channel apex. Such a symmetric pattern deviates substantially from

the asymmetric pattern found here and in observations (e.g., refs. 8,20). As noted by Wearing et al.[59], further complexity may arise through dependence on a variety of factors, including basal slope, ocean heat content, plume entrainment rates, along-flow position, Coriolis effects, and ice-cavity circulation. To this end, our 3D ISOBC model is a potential candidate for the ocean component to be coupled with existing ice sheet models. To highlight further the need to capture the detailed vertical structure of the channelized ISOBC in such a model, the influence of adopting a coarser (10 m vs 4 m) vertical resolution is examined in S3 (see Supplementary Note 2). Use of the coarser vertical grid results in poor resolution of the near-ice turbulence and meltwater-induced stratification, as well as the pycnocline structure inside shallower (60 and 120 m) channels, features that influence the assessment of channelized quantities.

Outflowing meltwater from beneath Antarctic ice shelves has become a topic of focus in recent years because of its important role, as sketched in the lower panel of Fig. 8, in coastal polynya formation[8,15–17], shelf water modification[18,19], shelf circulation variability[60,61], and sea ice growth/melting[62,63], all of which ultimately impact the global climate system[54–56]. Such influences exerted by basal meltwater outflows from beneath Antarctic ice shelves are evaluated using a variety of global coupled models in which, however, there are great uncertainties in the location and amounts of the applied basal meltwater influx[50–56]. Prominent basal channels propagating to the ice front are important conduits, delivering basal meltwater to the open seas. Therefore, the quantification of channelized mass, heat, and buoyancy discharge established here (Fig. 7c, d, f) has the potential to reduce these uncertainties, with the caveat that exploring the relationships between channelized quantities and channel CSS beyond the present CSS range represents a further important step. Although the relationships are derived for idealized configurations that undoubtedly differ from the realistic setting in important ways, including a wide range of basal conditions and ice shelf dynamics, they may provide preliminary estimates with credible magnitude.

This study demonstrates that deep channels may lead to substantial amplification in channelized basal melting, meltwater channeling, and plume warming and salinization (as illustrated in the upper panel of Fig. 8). In other words, there might be a transition to warmer and saltier near-ice hydrography if basal channels deepen, a transition that is attributed to the changes in thermodynamic processes and associated thermohaline structure inside the deepening channels (upper panel of Fig. 8). Therefore, there may be a pressing need for hydrographic and turbulence observations from within ice shelf basal channels, as well as further observations of evolving basal channel systems and hydrography adjacent to the basal channel outlets at ice fronts.

## Methods
### 3D ISOBC model
The 3D ISOBC model is developed here based on an earlier 2.5D vertical slice model[46], both of which incorporate a $k$-$\varepsilon$ turbulence closure. In the 2.5D model, despite the inclusion of the Coriolis force, the across-slope gradients of all variables are omitted, in contrast to the present 3D ISOBC model. The complete set of 3D governing equations of the ISOBC, including the mass, momentum, temperature, and salinity equations with a seawater-state equation, describes the 3D evolution of the dynamics and thermohaline processes of the boundary current adjacent to the ice shelf base under the predominant buoyancy forcing[43]. The adopted coordinate system is set as $x$-axis pointing upslope (positive), $y$-axis across the slope, and a $z$-axis perpendicular to the planar ice base, with its origin set at the channel apex. All the governing equations are discretized in finite-difference form, the Coriolis terms are treated using a semi-implicit scheme[64], the nonlinear terms are discretized using the Total Variation Diminishing (TVD)

schemes[64], and the nonhydrostatic pressure is solved using the Successive Over Relaxation (S.O.R.) method[64].

### Model configuration
The along-slope ($x$) extent of the domain is set to 60 km. In the across-slope ($y$) direction, the extent of the planar base adjacent to each side of the channel is set to 10 km, so the total across-slope extent is 20 km plus the specific channel width. In the $z$ direction (positive upward from the planar ice base), the extent underneath the planar part of the ice base is set to 60 m; within the channel area the channel height is added to the vertical extent of the domain. The grid resolution $dx = 750$ m, $dy = 400$ m, and $dz = 4$ m in the $x$, $y$, and $z$ directions, respectively. The southern (i.e., the deepest) boundary is set as closed but free-slip, except for the buoyant inflow section that represents an external GMW input from outside our domain to initiate the ISOBC. The inflow dimensions and location are kept the same in all runs and are set to 2 km in width by 28 m in height. The inflow is set immediately to the east of, rather than inside, the channel to avoid differences caused by the different channel CSS in S1 runs. The northern boundary is set as an open boundary with zero-gradients, while the western and eastern boundaries are reentrant so that periodic lateral boundaries enable us to form a consistent, permanent, and well-developed meltwater-cycling system within our domain. Thus, the role the basal channel plays in the ice–ocean interactions can be evaluated in a setting where it is the only conduit for geostrophic flow toward the ice front. Nevertheless, the existence of closed boundaries and the occasional proximity of channels to them are, in fact, features of real basal meltwater conduit systems, which, therefore, could be incorporated into future real-case studies.

The shear stress at the ice–ocean interface is parameterized using a quadratic drag law with drag coefficient $Cd = 0.0025$, and the corresponding scalar fluxes are calculated by the three-equation formulation[65]. The lower boundary is set to zero gradients.

The ocean domain is initially at rest. The inflow and ambient water properties are prescribed according to hydrographic conditions appropriate for Pine Island Ice Shelf. The ambient water has potential temperature $T = 0.18\,°C$ and salinity $S = 34.36$ psu, based on the simulated results of Dutrieux et al.[66], while the inflow has $T = -0.1\,°C$ and $S = 34.26$ psu, according to the mixing (Gade[57]) line observed during an Autonomous Underwater Vehicle (AUV) survey within the Pine Island Ice Shelf cavity[67]. For model stability both the inflow speed and thickness are ramped up from zero to $0.06\,m\,s^{-1}$ and 28 m, respectively, over the first 3 days, during which the scalar fluxes at the ice–ocean interface are set to zero. After that period, the calculation of the melt rate and the corresponding scalar fluxes are enabled.

The overall ice shelf draft decreases linearly from 870 m at the southern boundary to 420 m at the northern boundary (i.e., the calving front) with a moderate basal slope $J = 0.0075$. An inverted basal channel with width $W = 8$ km, height $H = 140$ m, and an idealized sine-shaped cross-section is centered in the domain, extending over the whole along-slope dimension (i.e., the reference CSS). The Coriolis parameter is set to $-1.41 \times 10^{-4}\,s^{-1}$. The time step is set to 10 s to satisfy computational cost and stability constraints. The time integration is set to 1800 h (75 days) after which the results of all the simulations conducted here have reached a steady state. Under the present model configuration, each simulation takes several days, depending on the selected channel CSS, because the 3D ISOBC model is not yet coded in parallel. The above settings are used in the reference run (i.e., H140W8), while sensitivity runs, having all parameters except the one under investigation held fixed, have been carried out and divided into three classes (see Table 1 for more details).

### Calculation of channelized outflowing transport
To determine the discharge and the associated transport quantities, we have to first define the upper buoyant plume thickness ($D$; m)

within the channel. We follow the definition of Arneborg et al.[68], which uses a buoyancy-weighted form as follows:

$$D = \frac{2\int_0^{H_w} g' z' dz'}{\int_0^{H_w} g' dz'} \qquad (1)$$

where $H_w$ is the water column thickness (m), $z'$ is the absolute value of $z$ (m), $g' = \frac{\rho_a - \rho}{\rho_0} g$ is the reduced gravity (m s$^{-2}$), $g$ is the gravitational acceleration (9.81 m s$^{-2}$), $\rho_a$ is the ambient water density (kg m$^{-3}$), $\rho$ is the local water density (kg m$^{-3}$), and $\rho_0$ is the reference water density (1030 kg m$^{-3}$). Based on the above definition, at the northern boundary, the channelized plume discharge $Q$ (Sv; $1\,\text{Sv} = 10^6\,\text{m}^3\,\text{s}^{-1}$), the associated heat discharge $Q_H$ (TW) and buoyancy discharge $Q_B$ (m$^4$ s$^{-3}$), and the discharge-averaged plume heat $\hat{H}$ (MJ m$^{-3}$) and buoyancy $\hat{B}$ (m s$^{-2}$) are calculated as follows:

$$Q = \int_{10\,km}^{10\,km+W} \int_0^D u\,dz'dy \times 10^{-6} \qquad (2)$$

$$Q_H = \rho_0 c_0 \int_{10\,km}^{10\,km+W} \int_0^D u\left(T - T_f^s\right) dz'dy \times 10^{-12} \qquad (3)$$

$$Q_B = \int_{10\,km}^{10\,km+W} \int_0^D u g' dz'dy \qquad (4)$$

$$\hat{H} = \frac{Q_H}{Q} \qquad (5)$$

$$\hat{B} = \frac{Q_B}{Q} \times 10^{-6} \qquad (6)$$

where $u$ is the along-slope velocity (m s$^{-1}$), $c_0$ is the specific heat capacity (3974 J kg$^{-1}$ °C$^{-1}$), and $T_f^s$ is the surface freezing point (−1.865 °C corresponding to a representative surface salinity of 34 psu for Pine Island Bay[66]). Positive $Q$ is out of the cavity.

## Data availability
The full set of simulation data in this study is available from the corresponding author upon request or can be directly generated by running the 3D ISOBC model, which is publicly available.

## Code availability
The original source codes for the 3D ISOBC model (V1.0) are available from https://github.com/cc2713206/3D-ISOBC-Model_NatCommun.

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

## Acknowledgements

This work was supported by the Independent Research Foundation of Southern Marine Science and Engineering Guangdong Laboratory (Zhuhai) (SML2023SP201) (C.C., Z.W., and C.L.), by the National Natural Science Foundation of China (41941007) (C.C., Z.W., and C.L.), by the Southern Marine Science and Engineering Guangdong Laboratory (Zhuhai) (313022001) (C.C.) (313022009) (Z.W.) (313021004) (C.L.), and

by the Natural Science Foundation of Jiangsu Province (BK20191405) (C.C.). The authors are grateful to Luyu Shen for developing and maintaining the source code of the 3D ISOBC model and to Keith W. Nicholls for helpful discussions while writing the paper.

## Author contributions

The study was conceptualized by C.C., A.J., and P.R.H. C.C. developed the 3D ISOBC model, devised and carried out the experiments, analyzed the output, and wrote the paper. A.J. and P.R.H. assisted with experimental design and output analysis. Z.W. was responsible for funding acquisition. A.J., P.R.H., Z.W., and J.D. assisted with paper revision. A.J., P.R.H., and C.L. assisted with paper writing. C.C., A.J., P.R.H., Z.W., J.D., and C.L. contributed to the development of ideas, provided feedback on the paper, and approved the final version of the paper.

## Competing interests

The authors declare no competing interests.
