## [Peer Review File · Nature Communications]

Ice shelf basal channel shape determines channelized ice-ocean interactionsREVIEWER COMMENTS

Reviewer #1 (Remarks to the Author):

Please see the attached document.

Reviewer #2 (Remarks to the Author):

I will upload the review report as an attachment.

Reviewer #3 (Remarks to the Author):

Overall comments:

The article describes the results from a new 3D ice shelf-ocean boundary current model, which represents vertical mixing in meltwater plumes beneath the ice shelf. The work is important due to the role of plumes and channels in ice shelf stability, and the work has the potential to aid our understanding of the processes that are going on. The findings over the influence of "cross-sectional shapes" on melt rates were noteworthy. The work has the potential to be of significance to both glaciology and oceanography communities.

So, in principle the work is worth publication, however, in the format it is in presently, the manuscript doesn't show off the work to the best of its ability. There is a lot of qualitative/descriptive words in the results sections, describing all the results in detail rather than drawing the reader to the important points. There were a lot of "appears to" statements, which needed to be more definite statements, and "relatively"s that needed quantifying. Some features of the figures were described in depth, an indication of these particular areas on the figures would be useful.

I was hopeful the discussion would draw the results together, and while there were some summary statements, there were also some areas of vagueness. For example, lines 392-396 make a big statement about profound impacts, but doesn't elaborate what the impacts are. Therefore, it is not so much that the work doesn't "support" the conclusions and claims, but it is that the conclusions and claims could be better connected to the results.

I think in summary, the paper has promise, the model looks great with a sound methodology, and there's some interesting results, but the paper needs a tighter focus. I'm not sure if it is a model description paper (that's what the results section feels like) or providing a clear result with important implications (what the discussion feels like it is trying to do). I was not much wiser at the end of the discussion about the advances made to the big picture topics outlined in the introduction. I think with a bit of a re-write to be more focussed, this could be a neat paper.

Figures:

The contour labels on plan figures (e.g. Fig.1) are impossible to read – these need to be much bigger. Why are g-i) smaller in Figure 4? Was the overall domain narrower as well as the channel width? The numbering system on Figs 5 & 6, 8 is a bit muddled and hard to follow.

Response to reviewers:

Ice shelf basal channel shape determines channelized ice-ocean interactions

Chen Cheng, Adrian Jenkins, Paul R. Holland, Zhaomin Wang, Jihai Dong, Chengyan Liu

We are grateful to Qin Zhou and other two anonymous reviewers for their very constructive and helpful comments and suggestions. In the following, *italic* denotes the reviewers' comments, and the following is our response. "l. #" and "L. #" denote line # in the previous and revised manuscripts, respectively. Double quote and angle bracket represent the excerpts of the previous and revised manuscripts, respectively. New references have been given the full citation in the end of this response. All the changes from the previous manuscript are shown in **blue** characters in the attached file 'manu2-r1-showing changes.pdf'.

Response to Reviewer #1:

The manuscript by Cheng et al., 2023 presents an 3D k-epsilon model of large scale ice-ocean basal channels of the scales reported below Antarctic ice shelves such as Pine Island. The primary contribution of this manuscript is to present the dependence of melt rate, heat and mass transport, and channel secondary overturning circulation as a function of the following parameters: channel width and height, shelf slope, ambient conditions and model vertical resolution. As such, this paper would make a useful and important contribution to ice shelf-ocean interactions and its role/representation in climate models.

We thank the reviewer for their support.

Overall Recommendation:

I have three primary concerns that I hope the authors might address: (1) the choice of y-(across-channel direction) periodicity for the modeling domain, (2) resolution limitations of the geometric endmember cases, and (3) the flat shelf case does not seem consistent with the empirical fits proposed in the parameter sensitivity results. These concerns are highlighted in the comments below.

These three primary concerns are all very constructive, so please refer to our response to each of them below.

Major Comments:

Lines 40-42: I find it a bit confusing here that you make the point that channelization may stabilize ice shelves, yet later (Lines 51-53), you say that channelized outflows “reduces the structural integrity of the ice shelves. Can you clarify the distinction of processes/scales over which channelization may be a stabilizer or destabilizer?”

Sorry for the statement made for the direct connection between the ice shelf stability and the changes in basal melting pattern caused by the existence of channels, only based on ref 12. Actually, that is essentially a complicated problem that has as yet not been satisfactorily answered. So, to eliminate the ambiguity here, we revise

“They directly influence ice-shelf stability by interacting with ice-shelf fractures, leading to along-channel retreat^{8,9} and frontal calving^{10,11}. As to the overall stability, however, the existence of channels stabilizes ice shelves by distributing melting across the ice shelf rather than concentrating it in one place¹².”_l. 39-42

as

<They directly influence ice-shelf stability by interacting with ice-shelf fractures, leading to along-channel retreat^{8,9} and frontal calving^{10,11}. **The existence of channels can also control the ice-shelf mass balance by distributing the basal melting across the ice shelf¹²⁻¹⁴.**>_l. 46-50.

Lines 77: Could you succinctly spell out what 2.5D refers to here so the reader does not need to refer to the Cheng et al. 2022 reference?

We revise

“Such an ISOBC model would represent a further development of models from 1D³⁴⁻³⁶ to 2.5D³⁷ that have been used to investigate the dynamic and thermodynamic vertical structures of the ISOBC.”_l. 76-78

as

<Such an ISOBC model would represent a further development of models from **vertical** 1D⁴³⁻⁴⁵ to 2.5D (**the across-slope gradients of all variables are omitted**) **vertical slice models**⁴⁶ that have been used to investigate the **fundamental** dynamic and thermodynamic vertical structures of the ISOBC.>_L. 97-100.

Lines 95: Could you be clearer here whether the meltwater buoyancy source overtakes the southern boundary GMW influx in time or proximity to the southern boundary?

We add a further explanation as

<All the simulations are forced by a constant GMW **buoyancy** influx (**2.3 m⁴ s⁻³**; **Methods**) along a stretch of the deepest (southern) boundary, although the basal melting over the whole domain becomes the predominant buoyancy source **quickly**. **In our reference run, for instance, the outflowing channelized buoyancy discharge (Methods) overtakes that GMW buoyancy influx within only 14 h after the channelized GMW plume first arrives at the northern boundary, and subsequently reaches a quasi-steady value of 296 m⁴ s⁻³.**>_L. 123-129.

Lines 78-81: This sentence is not clear and needs rewriting. I don't think it is reasonable to say that “these models suggest the existence of a near-ice frictional boundary layer” (this needs to be the case at any rigid fluid boundary?). Also this sentence seems to equate the near-ice frictional boundary layer to the plume region and the exterior geostrophic flow to the pycnocline whereas the near-ice frictional boundary layer should be a small interior thickness of the plume and the exterior pycnocline might be a transition to the exterior geostrophic flow? Note also that later in Line 221, you reference the quasi-geostrophic equilibrium of submarine canyons instead of just geostrophy.

We revise

“Such an ISOBC model would represent a further development of models from 1D³⁴⁻³⁶ to 2.5D³⁷ that have been used to investigate the dynamic and thermodynamic vertical structures of the ISOBC. Those models suggest the existence of a near-ice frictional boundary layer and an exterior stratified geostrophic flow, with the former corresponding to an upper weakly stratified layer, i.e., the plume region, and the latter an underlying pycnocline.”_l. 76-81

as

<Such an ISOBC model would represent a further development of models from **vertical**

1D⁴³⁻⁴⁵ to 2.5D (the across-slope gradients of all variables are omitted) vertical slice models⁴⁶ that have been used to investigate the fundamental dynamic and thermodynamic vertical structures of the ISOBC. Those models suggest an upper well-mixed, turbulent layer (the plume region), an intermediate pycnocline suppressing mixing, and an exterior stratified geostrophic flow. However, these low-dimensional ISOBC models cannot investigate the spatiotemporal influence of ice shelf basal channels on the dynamics and thermohaline structure of the ISOBC in an essential three-dimensional way.>_L. 97-105.

Line 116, Line 283: I suggest breaking up S2 and the section “Modulation by other factors” into two separate subsections: slope and ambient T/S properties – it seems clunky to combine these two parameter sensitivity discussions.

We divide S2 into S2-A and S2-S for the sensitivity to ambient properties and basal slope, respectively (see L. 148-150 and Table 1 in L. 755). The section “Modulation by other factors” is also broken up into the sections <Channelized quantities modulated by ambient properties> (L. 372) and <Channelized quantities modulated by basal slope> (L. 391).

Line 117, Lines 313-352: Does the vertical resolution section really belong with the other parameter sensitivity simulations? I think this currently reads more like supplemental information instead.

Line 313/Impact of vertical resolution: I would recommend moving this to the SI as it does in my opinion provide additional dynamical insight, but it is useful supporting information.

We move this part to the SI as Section S1 (L. 57 in SI), and add a summary as <To further demonstrate the need to capture the vertical structure of the channelized ISOBC in such a model, the influence of adopting a coarser (10 m vs 4 m) vertical resolution is examined in S3 (see Supplementary S1). It is found that the deficiency of vertical resolution results in poorly resolving the near-ice turbulence and meltwater-induced stratification, as well as the pycnocline structure inside shallower (60 and 120 m) channels, which jointly influence the assessment of channelized quantities.>_L. 429-435
in the section “Discussion”.

Line 127 and elsewhere: I suggest renaming the HOW8 case to just H0 for clarity.

Revised as you suggested.

Line 137 and HOW8, and reference case in general: This is my most significant concern for this generalizability of the modeling results from this study. Firstly, it is not explicitly mentioned in the text that the domain is periodic in y, but it seems that this must be the case based on Figure 1. In general, a presence of a cavity East and West boundary

would lead to a pressure gradient in y (whether or not these boundaries are within the range of y in your domain). Without this “western boundary current”, there is no/minimal outflow as seen in your HOW8 case. Would it make sense to test a case with either a non-periodic East and West boundary or pressure gradient in y . At minimum, this should be presented as a discussion point.

Thank you for this important point about the configuration of lateral boundary conditions. Actually, during the development of our 3D ISOBC model, we have tested the cases with closed western and eastern boundaries, and the former is shown to have more significant effects on the simulated results. Setting the western boundary as closed one, as you expected, we find that the meltwater piles up against the western boundary, establishing a pressure gradient in y , and outflows along this boundary as a “western boundary current”. However, that behavior strongly interferes our evaluation on the role basal channel plays in ice-ocean interactions. First, basal channel is no longer the sole conduit of meltwater outflow; the western boundary-ice front corner becomes the other main exit of meltwater, which significantly obscures the channeling function, i.e., the focus of this study. Second, with the decreasing distance from the western boundary to the channel, the piled meltwater inevitably extends into the channel, in this case the western part of the channel, flat shelf base, and the western boundary together can be regarded as constituting the irregular western part of a new channel. Thus, setting the western boundary as closed one would introduce another undesired uncertainty. Instead, setting the lateral boundaries as periodic ones, we can generate a consistent, permanent, and well-developed meltwater-cycling system which enables us to assess the interactions between the channel and the ISOBC adequately. So, we are sorry for that unclarity, and we add a necessary explanation on the adoption of periodic lateral boundaries in the subsection “Model configuration” as

<The northern boundary is set as an open boundary, with zero-gradients, while the western and eastern boundaries are reentrant, i.e., the periodic lateral boundaries. Actually, the closed western and eastern boundaries have been tested, and the former has more significant effects on the simulated results (not shown). The meltwater piles up against the western boundary, and outflows along this boundary as a “western boundary current”. Thus, the western boundary-ice front corner becomes the other main exit of meltwater. Moreover, the piled meltwater can also extend into the channel with the decreasing distance from the western boundary to the channel. To this end, setting the western and eastern boundaries to be periodic allows us to avoid these undesired uncertainties that significantly obscure the channeling function. In addition, the periodic lateral boundaries enable us to form a consistent, permanent, and well-developed meltwater-cycling system within our domain, so the role the basal channel plays in the ice-ocean interactions can be evaluated adequately.>_L. 507-520.

Line 159: Can you briefly explain here or in the discussion section something about the dynamics of the across-channel flow outside the channel (shown in Fig. 2 c,f,i)? Some features of the vertical variation in along-slope velocity and across-slope velocity looks to be suggestive of a poorly vertically-resolved Ekman boundary layer (with the k -

epsilon eddy viscosity parameterization). Is that the right characterization of the relationship of $u(z)$ and $v(z)$ outside the channel? I am also confused by the phrase “net across-channel flow discharge synchronously reduces”. Replace “due course” in Line 162 with an explicit callout to the section where this is further discussed.

Jenkins⁴³ first unraveled the fundamental vertical structure of ISOBC underneath a planar sloping ice shelf base (i.e., outside the channel here) (see his Fig. 3e, f): 1. the along-slope velocity u is purely ageostrophic, flowing to the right (in the Southern Hemisphere) of the geostrophic across-slope velocity v that has both ageostrophic and geostrophic components; 2. the geostrophic component of v is determined by vertical stratification, and the ageostrophic components of both u and v are identical to **the classical Ekman currents** but their magnitude is determined by the maximum in buoyancy forcing at the ice–ocean interface. As shown in Fig. 6b for H140W8, there is a prominent turbulent boundary layer of severalfold Ekman depth $d_E \approx 16$ m beneath the planar ice shelf base. Therefore, the present vertical resolution of 4 m would be adequate to resolve the Ekman boundary layer. However, because the nonlinear advection and horizontal diffusion terms were disregarded completely in the one-dimensional analytical analysis of Jenkins⁴³, the deviation of the characterization of relationship of u and v outside the channel from the analytical results of Jenkins⁴³ probably results from these terms. They are associated with the flow field nonuniformity which, as shown in Fig. 1d, may be attributed to the in-outflow variability along the open boundaries and the channel-induced topography nonuniformity. Hence, considering other two points suggested here, we revise “In addition, the net across-channel flow discharge synchronously reduces (Fig. 2c, f, i), and the manifestation of an anticlockwise secondary overturning becomes more apparent (as indicated by blue arrow in Fig. 2f, i), which is a critical dynamic feature discussed in depth in due course.”_1. 159-162

as

<With the continuous accumulation of meltwater from upstream of the channel along the slope (Fig. 1d), the across-slope GMW flow fills the channel (Fig. 2c), and eventually crosses it once the channel capacity is exceeded (Fig. 2f, i). Meanwhile, a weak eastward (negative) across-slope velocity (as indicated by blue arrow in Fig. 2i) underneath the western channel flank manifests a topographic secondary overturning, a critical dynamic feature discussed in depth in the next section. The deviation of the vertical structure of the ISOBC outside the channel from the fundamental structure presented in Jenkins⁴³ may be attributed to the lateral flow field nonuniformity shown in Fig. 1d.>_L. 199-207.

Lines 168-174. It might improve the discussion here to be more precise about the melt dependence on channel width, height, and aspect ratio. In particular, I think it is worth mentioning that H140 and H220 cases plateau beyond W8, while the sensitivity to channel width is still preserved between W4 and W8 except for the H60 case. I believe this is more accurate than the statement “dependence on channel width remarkably diminishes as channel shallows.” Similarly, for Fig. 3b, I think it is clearer to say that

the melt rate does not exhibit a strong trend (except the H300 case) for W8 and W12 (that it plateaus above W8) whereas all cases with W4 have proportionally lower melt rates?

To provide a more precise description of Fig. 3a and b, we revise

“The channelized basal melt rates, area-averaged over the channel domain, are very sensitive to changing channel CSS. On one side, the deepest (300 m) runs exhibit the relatively strongest increasing trend of channelized melt rate with channel width, but the dependence on channel width remarkably diminishes as channel shallows (Fig. 3a). On the other side, the melt rate clearly increases with channel height, but the rate of increase is greater in wider channels (Fig. 3b).”_l. 168-174

as

<The channelized basal melt rates, area-averaged over the channel domain, exhibits strong dependence on changing channel CSS. For the shallowest H60 cases, the channelized basal melt is insensitive to channel width, and is the weakest compared to other deeper cases (Fig. 3a). For H140 and H220 cases, the channelized basal melt increases with channel width but plateaus beyond W8. For the deepest H300 cases, the channelized basal melt is monotonically enhanced with the channel width. As shown in Fig. 3b, the increasing rate of channelized melt with channel height is larger for wider channels, but this increasing rate becomes less sensitive to channel width from W8 to W12, because H140 and H220 cases plateau beyond W8 (Fig. 3a).>_L. 213-222.

CSS section: I think the statement “the lower threshold of channel height ... is of no consequence to the ice-ocean interactions” does not seem well supported by these simulations. For instance, neither of the linear fits in Fig. 3b or 3c would fit the H0 flat case (e.g., the implicit claim that melt rate is insensitive to the channel widths or aspect ratios below those tested here), which I estimate to be about ~7 m/yr based on Fig. 1a. (It would be useful to provide a melt rate for the flat H0 case somewhere in this or the previous section.) I would include the H0 case to the Fig. 3 plots and edit the CSS section to reflect the fact that only this range of channel widths and heights were tested (due to resolution limitations) to avoid making the statement that there is no sensitivity outside of these parameter ranges (which the flat case would seemingly disprove).

Thank you for this important point that we completely agree with, and so we modify the statement

“Moreover, the regressions of melt rate onto aspect ratio have approximately the same intercept (Fig. 3c). It should be noted that we do not intend to further demarcate the lower threshold of channel height shallower than which the channel is of no consequence to the ice-ocean interactions.”_l. 176-180

with a necessary caveat statement

<Moreover, the regressions of melt rate onto aspect ratio have similar intercepts (Fig. 3c). In other words, the trend of channelized melt within the present CSS range does not fit the background value of 7.9 m yr⁻¹ in H0 (Fig.3 b, c). Consequently, owing to the model resolution limitation, a caveat is not avoidable, as the ranges of channel

height and width outside of the present 60 to 300 m and 4 to 12 km, respectively, have not been examined here. Accordingly, combining ... This quantification demonstrates that both channel height and aspect ratio regulate the overall magnitude of channelized basal melt, the applicability of which, however, needs to be extended for a wider CSS range.>_L. 225-236.

Line 204-205, Figure 5a: This panel/discussion seems a bit out of place as it is a spinup process, which is dependent on the choice of initial conditions?

We revise

“The evolution of discharge-averaged heat of channelized outflow (see Methods) for all S1 runs reveals a general melting-induced cooling over time, the level of which is modulated by the channel CSS (Fig. 5a).”_l. 204-206

as

<The discharge-averaged heat of channelized outflow (see Methods) for all S1 runs manifests an initial melting-induced cooling period (not shown), and progressively approaches the respective quasi-steady values which are modulated by the channel CSS (Fig. 5a).>_L. 263-266,

and remove the spinup period before 1080 h in Fig. 5a.

Line 246: This sentence implicitly makes the claim that the along-channel velocity is geostrophic (which may be true), but it is certainly possible that the plume can be ageostrophic (and especially likely while it is spinning up near its source). Can you verify that this flow is geostrophic in your simulations and state that the flow within the channel is primarily geostrophic (or comment on the along-channel transition to geostrophic balance)?

The underlying geostrophy of along-slope velocity inside the channel is quite well illustrated in Fig. 2. The isopycnals beneath the western part of channel evolve along the channel to become more tilted (Fig. 2a, d, g). The corresponding increase in the across-slope pressure gradient drives a collocated reinforced along-slope velocity (Fig. 2b, e, h). Accordingly, we add the following statement in the discussion of Fig. 2 as

<In detail, an ISOBC core leaning on the Coriolis-favored side of the channel gradually emerges, with increasingly larger buoyancy (Fig. 2a, d, g), more downward tilted isopycnals (Fig. 2a, d, g), and faster along-slope velocity (Fig. 2b, e, h). Accordingly, the geostrophy of along-slope velocity inside the channel is verified here: an increase in the across-slope pressure gradient corresponding to the gradually tilted isopycnals beneath the western part of channel accompanies the collocated stronger along-slope velocity.>_L. 192-199.

Line 231-234: This is a similar issue to the CSS section comment above. For the H60 case (only 15 gridpoints tall), vertical diffusion is likely much stronger due to mixing at the gridscale. I would just be more direct with these known caveats and/or careful in the way this is presented to the reader or conduct additional simulations at higher

resolution (not lower resolution like the ones in the resolution section).

To address concerns about model resolution influencing these results, we conduct 6 additional runs for the shallowest (H60) and narrowest (W4) cases, including H60W8 and H60W12 with $dz=2$ m, H60W4 with both $dz=2$ m and $dy=250$ m, and H140W4, H220W4, and H300W4 with $dy=250$ m. Please refer to our response below to your comments ‘CSS section: I think a possible caveat here ... w.r.t. the 4m resolution simulation’ for more details about the model resolution issue. In addition, to simultaneously resolve the specific concern of Reviewer #2, we discuss more in depth the thermodynamics inside the channel as elaborated in L. 293-336.

Line 248: Ekman transport directed to the left?

This Ekman transport is directed to the right of the geostrophic flow, as in the classical ISOBC result, taking into account the inverted ice shelf base and the basal stress in the opposite direction of the main channel flow. Nevertheless, we revise “near-ice Ekman transport directed to the right (Southern Hemisphere) of the main channel flow” _l. 247-248 as <near-ice Ekman transport directed to the east (Southern Hemisphere) of this main channel flow> _L. 318-320.

Line 254: What is the line of reasoning for fitting a log-law in Fig. 5a?

We revise “The predominant dependence of discharge-averaged channelized outflow heat on basal channel height (Fig. 5a) appears to be described by a log-law for the intermediate-width runs (blue component in Fig. 7a).” _l. 253-255 with the reasoning for fitting a log-law in now Fig. 8a as <The discharge-averaged channelized outflow heat is illustrated here to depend largely on basal channel height in comparison to channel width (Fig. 5a). That dependence for the intermediate-width runs can be simply described by a log-law, allowing for both a concise form and its more desirable performance than a power law (blue component in Fig. 8a).> _L. 337-341.

Figure 7: As earlier, these plots should include the flat H0 case as well. I think this would help lend some additional insight to the choices of empirical fits in Lines 254-274, which are currently rather opaque.

We agree with this point, but the no-channel case will correspond to no channelized discharge of any outflow quantities. In agreement with this, our empirical discharge p.u.w. power-law function of channel height (as labeled in Fig. 8a) returns to zero for the no-channel case. Nevertheless, for more clarity, we reset both y axes (i.e., discharge-averaged heat and discharge p.u.w.), as well as the x axis (i.e., channel

height) of Fig. 8a to zero origins. Again, as our response to your aforementioned comment, we reemphasize at the end of CSS section that

<Again, an imperative effort needs to be made to extend the applicability of all these empirical expressions of channelized meltwater discharge beyond the present range of CSS.>_L. 369-371.

Line 258: I'm not sure I understand the meaning of the sentence "that relation is found to be implicit, and is thus difficult to characterize."

We revise

"Accordingly, quantifying the heat discharge of channelized outflow (Methods) would be straightforward, if we know the relation between the outflowing discharge (Methods) and the channel CSS. However, that relation is found to be implicit, and is thus difficult to characterise. Even so, the channelized outflowing discharge p.u.w. (per unit width) reflects clear correlation with channel CSS (Fig. 7b); ..." _l. 255-260

as

<Accordingly, quantifying the heat discharge of channelized outflow (Methods) would be straightforward, if we know the relation between the outflowing discharge (Methods) and the channel CSS. However, that relation is **opaque, that is, the quantitative dependence of the outflowing discharge on the channel CSS is difficult to find directly.** Even so, the channelized outflowing discharge p.u.w. (per unit width) reflects **a clear correlation with channel CSS (Fig. 8b); ...>_L. 341-347.**

CSS section: I think a possible caveat here is that the W8 channel is only 20 gridpoints across, and the H60 channel is only 15 gridpoints tall, so the lack of sensitivity to these two channels dimensions below this number of gridpoints is not surprising, but might be artificially influenced by numerical gridscale diffusivity at this model resolution.

Line 313/Impact of vertical resolution: Like in the earlier comment, a vertical resolution higher than 4m if feasible would be useful to test convergence of the H60 channel (similarly, a horizontal resolution sensitivity test with horizontal resolution $dy=200m$). If this is overly expensive, consider e.g., interpolating the 4m QSS to a 2m vertical grid and spinning up for a shorter period of time to test/measure drift of the higher resolution simulation w.r.t. the 4m resolution simulation.

This is another important point that we totally agree with. To this end, we carry out 6 additional runs, that is, H60W8 and H60W12 with $dz=2$ m, H60W4 with both $dz=2$ m and $dy=250$ m, and H140W4, H220W4, and H300W4 with $dy=250$ m, to test any changes of the higher resolution simulations w.r.t. their default ones. Please note that constrained by our available computational resources, we have to adopt $dy=250$ m instead of 200 m you suggested. We list the relevant statistic data in Supplementary Table 1 (see L. 119 in SI), and add the following discussion in the section "Discussion": <However, considering the relatively poor representation of these channel CSS using the current resolution ($dy=400$ m, $dz=4$ m), it is worthwhile to examine the reliability of quantified relations between channelized quantities and channel CSS proposed here

by performing higher resolution ($dy=250$ m, $dz=2$ m) simulations for the shallowest H60 and the narrowest W4 cases. Thereupon, six additional runs including H60W8 and H60W12 with $dz=2$ m, H60W4 with both $dz=2$ m and $dy=250$ m, and H140W4, H220W4, and H300W4 with $dy=250$ m are conducted, and the statistics of the simulated channelized quantities are summarized in Supplementary Table 1. For W4 cases excluding H60W4, when the across-slope resolution increases by 60%, the maximal deviations of the channelized melt rate and transport quantities in the higher resolution are only 11% and 21%, respectively. The deviation of H60 cases with wider channels is even less, except for H60W4 that has the largest deviation among all these six runs; the corresponding deviations of the channelized melt rate, discharge, heat discharge, and buoyancy discharge are 22.7%, 30%, 40.7%, and 23.7%, respectively. Consequently, applying the empirical quantitative relations developed here to the basal channels close to the lower limit of cross-sectional area is likely untenable. It is thus worth, as stressed above, exploring the quantified relations between channelized quantities and channel CSS beyond the present CSS range.>_L. 448-466.

Lines 412-422: Can you be more direct/clear here whether the 3D ISOBC model you are using in this study also uses a k -epsilon turbulent closure? I made the assumption that it did, but it is not clear. Could you provide more details on the key differences between the 2.5D and 3D model? Is “Further numerical details...” only referring to the S.O.R. method or other methods as well (be specific if possible)?

We revise

“The new 3D ISOBC model is developed here based on an earlier 2.5D model incorporating a k - ϵ turbulence closure³⁷.”_L. 413-414

as

<The new 3D ISOBC model is developed here based on an earlier 2.5D vertical slice model⁴⁶, both of which incorporate a k - ϵ turbulence closure. In the 2.5D model, despite the inclusion of the Coriolis force, the across-slope gradients of all variables are omitted, in contrast with the present 3D ISOBC model.>_L. 481-484, and

“Further numerical details are provided by Kämpf⁶¹.”_L. 422

as

<Further details about the 3D discretization and all the associated numerical methodology are provided by Kämpf⁶⁴>_L. 492-494.

Figure 2. This may vary by reader preference, but this cross-section is looking southward instead of northward (into the page) or analogously has East on the left and West on the right in each of the panels. Suggest flipping the y-axes or reminding the reader of the East/West directions in the panels.

We indicate the East/West directions in the panels of all the cross-sectional figures.

Figure 3. Suggest editing y-axes label to “Melt dependence on aspect ratio” or similar.

Revised as suggested.

Minor Comments:

Line 18: it is unclear here whether “which are also modulated by channel width to different extent” refers to “channelized basal melting, meltwater channeling, and warming and salinization of the channel flow” or only “warming and salinization of the channel flow”. Would it also be more direct to specify that it is a lesser extent for all these three outputs?

Agree, but to be more accurate, we revise

“..., which are also modulated by channel width to different extent.”_l. 18

as

<All of these channelized quantities are also modulated by channel width, with the level of modulation determined by channel height.>_L. 21-22.

Line 22: Is “elusive” the best word for this process? Misspelling of “overturing”.

Line 23: “key” -> key ingredient? Is mixing the right word? There can be a secondary circulation without mixing, and strictly speaking the model used here would not resolve the vertical mixing from the overturning circulation, correct?

Because we amend the interpretation of the thermodynamic mechanism in deep channel (L. 293-336), in the abstract we revise

“The topographic secondary overturning, an elusive process disregarded in existing depth-integrated plume models, is revealed to be the key for the mixing inside deep channels.”_l. 22-24

as

<The complicated topographic overturning circulations are revealed to be responsible for the unique thermohaline structure inside deep channels.>_L. 26-27.

Line 32: “the inland” -> “inland ice” or similar?

“the inland” -> <the inland ice sheet>

Line 37: Does ocean-sourced here refer to open ocean-sourced? Could you also clarify the distinction between subglacially-sourced and grounding line-sourced?

We add a statement of the distinction among these three types of ice shelf basal channel as

<Basal channels that are ocean-sourced, subglacially sourced, and grounding line-sourced are widespread on the Antarctic ice shelves^{7,8}. The formation of ocean-sourced (not intersecting the grounding line) and grounding line-sourced channels is controlled by oceanic processes within ice cavities, and these two types do not correspond to subglacial outflows. Subglacially sourced channels are in contrast formed by subglacial

freshwater discharge at the grounding line.>_L. 41-46.

Line 55: Would it be more direct to edit to: “Ice shelf basal roughness via topographic features that span a range of lengthscales...”

Revised as you suggested.

Line 73: Does “upper and lower edges” mean the upper (ice-ocean) and lower (entrainment) boundary of the plumes?

Revised as you suggested.

Lines 118-124: Does the time in days need to be converted to hours in this paragraph?

Revised as you suggested.

Line 154: Suggest some editing of this sentence for clarity... e.g., spatially evolves along the channel... from the adjustment, developing, and well-developed East-West (y-z) transects.

We revise

“In principle, the vertical structure of the ISOBC experiences consecutively the adjustment (Fig. 2a-c), developing (Fig. 2d-f), and well-developed (Fig. 2g-i) regimes, from the deepest part to the northern end of the domain.”_l. 153-156

as

<In principle, the vertical structure of the ISOBC **spatially evolves along the channel, consecutively represented by** the adjustment (Fig. 2a-c), developing (Fig. 2d-f), and well-developed (Fig. 2g-i) **East-West (y-z) transects**, from the deepest part to the northern end of the domain.>_L. 189-192.

Lines 165: I suggest spelling out the acronym for “CSS” even if it is defined earlier as it is a section title.

Revised as you suggested.

Line 169: “On one side” -> “On one hand”? Similarly in Line 172.

Line 171: Maybe edit to “there is weak dependence on channel width for a channel height of 60 m.”

These sentences have been revised. Please refer to our response to your specific comment above.

Line 250: “overturning” misspelling.

For a new mechanism interpretation (L. 293-336), this sentence has been removed.

Line 214: “much flat and sharp” -> “flatter and sharper”?

We revise

“The upper interface of the underlying thermocline is curved caused by the channel geometry, but its lower interface is relatively much flat and sharp.”_l. 213-214

as

<The upper interface of the underlying thermocline is curved **into the channel**, caused by the channel geometry, but its lower interface is, **in contrast, flatter and vertically sharper.**>_L. 276-279.

Line 259: reflects “a” clear correlation?

Corrected

Line 302: monotonously -> monotonically?

Revised

Table 1: Could you bold the reference case “H140W8”? Is there a way to emphasize/clarify that the H140W8 is repeated in S1 and S2 or simply combine it into S1? Might it make sense to rename H0W8 to just H0 since the W8 is a bit misleading?

The reference case H140W8 has been separately listed in the first row of Table 1 (L. 755), and H0W8 has been renamed to H0 throughout the manuscript.

Response to Reviewer #2 (Qin Zhou):

General Comments:

In this study, the authors explore the impact of basal channel shapes on ice-ocean interactions using a newly developed 3D Ice Shelf-Ocean Boundary Current (ISOBC) model. By effectively capturing vertical thermal and dynamic structure within the channel, their findings reveal that deeper channels notably amplify channelized basal melting, meltwater channeling, and warming. Moreover, a topographic secondary overturning in deeper channels that are not represented in existing depth-integrated plume models is identified by their model results. The authors state that secondary overturning is vital for the mixing inside deep channels. Because of the importance of studying ice-ocean interactions in the basal channels for understanding future ice-shelf stability and its wider implications on sea-level rise, I think this paper is interesting, important, and worth publication.

We thank the reviewer for their support.

However, I do have several concerns that should be addressed before I consider it ready for publication in Nature Communications.

1. In reviewing this manuscript, I've noted several instances where mechanisms, analogies, and the logical progression from detailed findings to conclusions might benefit from further clarity and explicit elaboration. This overarching concern centers on ensuring the findings are comprehensible to a broader audience, including readers who may be less familiar with the specific details of the ISOBC model or specific terminologies used.

We totally agree with the reviewer on this overarching point, and believe that the clarity and explication of this manuscript is significantly improved based on your detailed comments listed below.

For instance:

• The concepts of 'geostrophic adjustment' in line 139 and 'baroclinic response' in line 239 are introduced but not explained in detail.

We revise

“The ISOBC within the channel undergoes the geostrophic adjustment beneath the deepest part of ice shelf, and ...”_1. 138-139

as

<The ISOBC within the channel undergoes geostrophic adjustment, where the flow is governed by a balance between Coriolis force and the pressure gradient caused by isopycnals steered by the topography of basal channel, beneath the deepest part of ice shelf, and ...>_L. 172-175, and revise

“Its generation can be regarded as a baroclinic response to the increasing stretching of isopycnals (Fig. 5b-e and Supplementary Fig. 2a-d) induced by the channel deepening.”_l. 239-241

as

<the occurrence of this recirculation can be regarded as a dynamic response to a larger across-channel pressure gradient induced by more tilted isopycnals and steeper topography (Supplementary Fig. 2b) than that in the shallow (60 m) channel (Supplementary Fig. 2a).>_L. 313-317.

• *The analogy between pinched isopycnals in the deeper channels and the distribution of dense water in submarine canyons (lines 219-221) could benefit from a clearer contextual explanation. Specifically, how does this analogy provide insights or context into the processes described in the basal channel?*

The meltwater plume in the inverted basal channels seems to be analogous to the dense current in submarine canyons, but there is a critical difference between them: the ice shelf base is a reactive boundary, and the properties of the ISOBC are set by its interaction with this boundary, providing a continuous stabilizing buoyancy flux. Besides this difference, in Wåhlin⁴¹ the ambient water overlying the dense water is assumed to be stagnant, and no exchange between these two waters is considered, which is significantly different from the situation considered in this study. Moreover, the dense current is conventionally set to be supplied by the individual upstream source(s), while as shown in this study, the whole ice shelf base eventually provides the buoyancy for the global ISOBC. Therefore, based on the review of these differences inspired by your concern, the analytical bulk distribution of the dense water in submarine canyons from Wåhlin⁴¹ may be inapplicable to the ISOBC within ice shelf basal channels, and that analogy is thus removed here. Instead, as we respond to your subsequent comment ‘3. *The manuscript consistently underscores the critical role of secondary overturning in enhancing vertical mixing, ...*’, we propose a new mechanism for the generation of the pinched isopycnals in the deeper channels.

• *There are statements like the one concluding that the mixing of GMW with ambient warm water is enhanced in deep basal channels (lines 221-223), where the connection between the detailed findings and the conclusion could be made more explicit.*

Thank you for this important point. First, we revise the description of variability of the vertical thermal structure induced by the deepening channel shown in Fig. 5b to e as <...>, we find that a fully-developed relatively cold (lower than -1.4 °C) mixed layer builds up underneath the ice base across the full-width channel when the basal channel is shallow (Fig. 5b). The upper interface of the underlying thermocline is curved into the channel, caused by the channel geometry, but its lower interface is, in contrast, flatter and vertically sharper. However, when the basal channel deepens to 140 m, the upper mixed layer becomes warmer, and its across-channel coherence breaks off (Fig. 5c). The upper interface of the thermocline intersects the base immediately east of the

channel apex, and its lower interface has a weaker stratification, intruding upwards into the channel. When the channel further deepens (Fig. 5d, e), the meltwater plume east of the channel apex becomes even warmer. The upper relatively cold meltwater on the other side dwindles significantly, only confined in the top of the channel with the underlying isotherms pinched in the top-east of the channel.>_L. 274-286.

Second, based on your subsequent comment ‘3. *The manuscript consistently underscores the critical role of secondary overturning in enhancing vertical mixing, ...*’, we reconsider the hydrodynamic changes caused by the channel deepening in L. 293-336 to explain the associated thermohaline changes mentioned above, and consequently find that the statement of “the mixing of GMW with the ambient warm water is substantially enhanced within the deep basal channels” is not accurate enough. Thus, that statement has been removed here.

- *‘Overflow’ doesn’t seem to be a recurrent term in studies related to buoyant plumes caused by basal melting. If the authors are using the term in a novel context, additional explanation is required.*

In line 141, the term ‘overflow’ needs to be clarified, see the above general comments.

We remove the term “overflow”, and revise

“the ISOBC’s overflow across the channel western boundary appears to ...”_l. 141-142
as

<the GMW flowing across the channel western boundary becomes ...>_L. 177. The sentence

“This Ekman transport and the opposing overflow from the upstream of the channel conspire to ...”_l. 248-249

is removed because of a new mechanism interpretation as we respond later.

- *I suggest the authors clarify the relationship between the channel width and other variables in the text, like ‘... modulated by channel width to different extent’ in the abstract and ‘modulated by channel width’ in line 261, to ensure the readers fully understand the findings.*

Agree. We revise

“..., which are also modulated by channel width to different extent.”_l. 18
as

<All of these channelized quantities are also modulated by channel width, with the level of modulation determined by channel height.>_L. 21-22, and

“Even so, the channelized outflowing discharge p.u.w. (per unit width) reflects clear correlation with channel CSS (Fig. 7b); its magnitude increases with channel height, but is modulated by channel width.”_l. 259-261

as

<Even so, the channelized outflowing discharge p.u.w. (per unit width) reflects a clear

correlation with channel CSS (Fig. 8b); its magnitude increases with channel height, but is modulated by channel width, that is, the variability induced by channel width increases with channel height.>_L.345-348.

I encourage the authors to carefully go through the manuscript and address such areas of potential ambiguity, ensuring each mechanism or analogy introduced is sufficiently explained and its relevance made clear.

The potential ambiguity through the manuscript has been eliminated as carefully as possible.

2. The mechanism driving the key result, the secondary overturning, requires a more detailed explanation than currently presented in lines 243-250. To improve clarity around these critical findings, I propose the incorporation of a schematic or illustration detailing the dynamics behind the secondary overturning circulation, such as the direction of Ekman transport and the so-called 'overflow'. An exemplary format could be Figure 1 in Walin's (2004) paper, given the citation to this work in the present study. Moreover, it would be beneficial to elucidate the primary overturning mechanism—specifically, the underlying reasons for the existence of overturning within the channel in the first place. Additionally, does there exist a critical aspect ratio (or any other dimensionless number) of the channel that determines the generation or suppression of the secondary overturning? Identifying such a critical value could offer a deeper understanding of the conditions under which this mechanism operates and its broader implications.

Thank you for this important point. First, we add a schematic (Fig. 7; L. 803) summarizing the changes in both stratification and hydrodynamics caused by varying channel height. As you see in this figure, we amend the predominant interpretation, and highlight the underlying major mechanisms for the generation of the complicated thermodynamics in deep channel. Please refer to our response to your next comment for more details. In addition, there always exists an Ekman transport directed to the right of the main flow underneath the western part of the channel, this transport is merely so weak in shallow channels that it is hard to see in the figures. As the channel gradually deepens, the Ekman transport becomes more prominent. Therefore, this is a continuously evolving process, and it would thus be hard to demarcate a definite aspect ratio to identify whether the overturning is strong or weak. Nevertheless, as you see in Fig. 6f, the eastward Ekman transport emerges to be discernable in H140W8 when the channel deepens incrementally.

Figure 7. Schematic comparison of stratification and hydrodynamics within shallow and deep channels. Shallow channels (left) have little influence on the properties of the across-channel GMW flow. Little GMW is recirculated upwards along the western channel flank. In contrast, the stratification and hydrodynamics are significantly changed in deep channels (right) by four critical coherent processes as illustrated. The arrow colour transitioning from blue to pinkish red denotes the transformation of GMW to warmer iGMW, and vice versa. Not to scale.

3. The manuscript consistently underscores the critical role of secondary overturning in enhancing vertical mixing, as mentioned both in the abstract and lines 250-252. Contrarily, the authors also emphasize the prominence of vertical velocity over vertical diffusion in influencing full-channel-depth mixing. This position seems counterintuitive, especially given the conventional oceanographic understanding that regions of strong mixing are typically associated with elevated vertical diffusivity. Furthermore, the model results indicate that regions of pronounced secondary overturning correspond to areas with diminished vertical diffusivity, as described by the authors in lines 231-233. An additional point of contention arises in the S3 runs, which show stronger vertical diffusion on the channel's western side despite their coarser vertical resolution (Figure 8). If this coarser grid fails to capture the sharp gradient of the isopycnals, it could lead to the gradient being effectively smoothed, potentially resulting in an underestimation of vertical mixing rather than an enhancement.

These are important comments that we totally agree with. After carefully considering this point, we acknowledge that the previous mechanism interpretation is not fully consistent with the results shown in the corresponding figures. To this end, we amend the thermodynamic analysis here as

<All these thermohaline changes are closely tied to the hydrodynamic changes within the channels as they are deepened. Two key thermodynamic processes, vertical diffusion and vertical advection (Fig. 6), are considered. The distribution of

parameterized vertical diffusivity and resolved vertical velocity in two across-channel sections, i.e., $x=30$ and 60 km, within the well-developed regime are averaged for the following analysis. When the basal channel is shallow (i.e., in H60W8), the vertical diffusion is strong everywhere beneath the ice (Fig. 6a), which is consistent with the most well-defined upper mixed layer shown in Supplementary Fig. 2a. In addition, considering the vertical velocity, the positive-to-negative symmetrical pattern within the mixed layer suggests a flow passing across the shallow channel (Fig. 6e).

However, when the channel is deepened to 140 m (i.e., in H140W8), the eastern channel flank lengthens and steepens, which leads to a pronounced mixing between the GMW inflow rising more quickly along the eastern flank and the ambient water (Supplementary Fig. 2b), characterized by a large vertical diffusivity therein (Fig. 6b). That mixing substantially warms (Fig. 5c) and salinizes (Supplementary Fig. 1b) the GMW, so the resultant water mass is referred to as intermediate GMW (iGMW) here, and generates the aforementioned deformed pycnocline structure (Supplementary Fig. 2b). Although most of the iGMW crosses the channel (Fig. 2f, i), some iGMW is recirculated by a weak eastward Ekman transport adjacent to the western channel flank (Figs. 2i, 6f). Since the geostrophy of along-slope velocity inside the channel has been verified in Fig. 2, the occurrence of this recirculation can be regarded as a dynamic response to a larger across-channel pressure gradient induced by more tilted isopycnals and steeper topography (Supplementary Fig. 2b) than that in the shallow (60 m) channel (Supplementary Fig. 2a). In other words, larger across-channel pressure gradient results in larger geostrophic along-channel flow that leads to an increasingly significant near-ice Ekman transport directed to the east (Southern Hemisphere) of this main channel flow. The recirculation of iGMW upward along the western channel flank makes the water, interacting with the ice base, warmer (Fig. 5c) and saltier (Supplementary Fig. 1b) than that in H60W8 (Fig. 5b and Supplementary Fig. 1a). This recirculation, combined with the faster main channel flow, leads to higher basal melting of the western channel flank. The resultant strong stratification (Supplementary Fig. 2b) thus suppresses local turbulent diffusion substantially (Fig. 6b).

When the channel is deepened further (i.e., in H220W8 and H300W8), the coherent processes discussed above are displayed more prominently in the mixing between GMW and ambient water along the eastern channel flank (Fig. 6c, d), the recirculation of even warmer (Fig. 5d, e) and saltier (Supplementary Fig. 1c, d) iGMW upward along the western channel flank (Fig. 6g, h), and the resultant diminished vertical diffusivity (Fig. 6c, d). Furthermore, the progressive recession of lighter GMW and the advance of denser iGMW conspire to form the unique pinched isopycnals (Supplementary Fig. 2c, d). These complicated thermodynamic mechanisms responsible for the changes in the thermohaline structure with the deepening of basal channels are summarized in Fig. 7. >_L. 293-336.

Accordingly, in the abstract we revise

“The topographic secondary overturning, an elusive process disregarded in existing depth-integrated plume models, is revealed to be the key for the mixing inside deep channels.”_L. 22-24

as

<The complicated topographic overturning circulations are revealed to be responsible for the unique thermohaline structure inside deep channels.>_L. 26-27.

In addition, the reason why the turbulent diffusivity is markedly suppressed beneath the western channel flank is probably the prominent stratification along the recirculation of relatively warm and salty iGMW. Therefore, as shown in now Supplementary Fig. 2 for channel height larger than 60 m, this stratification is artificially relieved for $dz=10$ m, which results in the corresponding overestimated vertical diffusivity shown in now Supplementary Fig. 8. To this end, we add the relevant discussion in now Supplementary S1 as

<Moreover, it can also be found that the prominent stratification along the recirculation of iGMW (Supplementary Fig. 2b) is markedly relieved in H140W8-C (Supplementary Fig. 2f), which results in the corresponding overestimated vertical diffusivity along the western channel flank (Supplementary Fig. 8b).>_L. 94-97 in SI.

4. Another recurrent issue noted in the manuscript is the misalignment between figure descriptions in the text and the visual content of the figures themselves. This discrepancy occasionally makes it challenging to follow the arguments or extract the expected takeaways from the figures. Here are a few instances where this inconsistency is evident:

- 'Net across-channel flow discharge reduces' in line 159. It cannot be understood and it requires definition. From Figure 2c,f, i, I only see that the across-slope velocity increases in the channel. The authors may need to relate the across-slope flow discharge. I don't see 'anticlockwise secondary overturning' either; the arrows only show a place with weak or nearly zero across-slope velocity.

We remove this confusing phrase, and revise

"In addition, the net across-channel flow discharge synchronously reduces (Fig. 2c, f, i), and the manifestation of an anticlockwise secondary overturning becomes more apparent (as indicated by blue arrow in Fig. 2f, i), which is a critical dynamic feature discussed in depth in due course."_l. 159-162

as

<With the continuous accumulation of meltwater from upstream of the channel along the slope (Fig. 1d), the across-slope GMW flow fills the channel (Fig. 2c), and eventually crosses it once the channel capacity is exceeded (Fig. 2f, i). Meanwhile, a weak eastward (negative) across-slope velocity (as indicated by blue arrow in Fig. 2i) underneath the western channel flank manifests a topographic secondary overturning, a critical dynamic feature discussed in depth in the next section.>_L. 199-205.

- The assertion in lines 207-208, 'channel height dominates the heat content of channelized outflow, whereas channel width is of very little consequence', may require further clarification. From Figure 5a, it appears that the channel with a height of 300m and a width of 12 km exhibits a heat that aligns more closely with channels of 220m height rather than other channels of the same height. This suggests that channel width has a more significant role than suggested in the text, or there might be other factors at play. Further insight or clarification on this point is needed.

To be more accurate, we revise

“More specifically, channel height dominates the heat content of channelized outflow, whereas channel width is of very little consequence.”_L. 206-208

as

<More specifically, channel height dominates the heat content of channelized outflow, the level of which, however, diminishes as channel deepens. In contrast, the influence of channel width is rather limited; for the 300 m deep channel the variability of channelized heat caused by varying channel width is relatively largest, but only ranges from 3.6% for 4 km channel width to -4.7% for 12 km width relative to 8 km width.>_L. 266-271.

• *The study draws an association between Fig. 6f-h and Fig. 2f,i (lines 237-239), but the rationale for this connection isn't immediately clear. Specifically, Fig. 6f-h present vertical velocities averaged between $x=30\text{km}$ and $x=60\text{ km}$ for simulations H140W8, H220W8, and H300W8, respectively, but Fig.2f depicts across-slope velocities for sections $x=30\text{ km}$ and $x=60\text{ km}$ from the H140W8 simulation. Moreover, the secondary overturning is seemingly absent in H140W8 (Fig. 6f), making it difficult to correlate with Fig.2f,i.*

First, the blue arrow is now only indicated in Fig. 2i, as we respond above, to introduce the existence of a topographic secondary overturning.

<Meanwhile, a weak eastward (negative) across-slope velocity (as indicated by blue arrow in Fig. 2i) underneath the western channel flank manifests a topographic secondary overturning, a critical dynamic feature discussed in depth in the next section.>_L. 202-205

The reason behind that dynamic feature is subsequently detailed in our amended interpretation elucidated above wherein the association between Fig. 6 and Fig. 2 is revised as

<Although most of the iGMW crosses the channel (Fig. 2f, i), some iGMW is recirculated by a weak eastward Ekman transport adjacent to the western channel flank (Figs. 2i, 6f). Since the geostrophy of along-slope velocity inside the channel has been verified in Fig. 2, the occurrence of this recirculation can be regarded as a dynamic response to a larger across-channel pressure gradient induced by more tilted isopycnals and steeper topography (Supplementary Fig. 2b) than that in the shallow (60 m) channel (Supplementary Fig. 2a).>_L. 310-317.

In addition, ‘the secondary overturning is seemingly absent in H140W8 (Fig. 6f)’ exactly means that, as we respond above, the eastward Ekman transport gradually becomes more discernable in H140W8 when the channel is deepened incrementally.

• *The statement in lines 216-217 that ‘when the basal channel deepens, the upper mixed layer progressively degenerates; it becomes warmer and focused on the western side of the channel.’. Based on the sentence structure, the pronoun ‘it’ seems to refer to the ‘upper mixed layer.’ However, upon examining Fig. 5, it appears that there isn't a homogeneous temperature layer that would represent a mixed layer in panels d and e.*

The isothermal lines either tilt along the western side of the channel or are vertically distributed near the channel's apex.

The description of Fig. 5 has been revised. Please refer to our response to your preceding comment ‘• There are statements like the one concluding that the mixing of GMW with ambient warm water is enhanced in deep basal channels (lines 221-223), where the connection between the detailed findings and the conclusion could be made more explicit.’.

I recommend the authors conduct a thorough review of the manuscript to ensure that all figure descriptions correspond accurately to their respective figures, ensuring clarity and cohesion for readers.

The misalignment between figure descriptions and their respective figures through the manuscript has been rectified as carefully as possible.

5. While I appreciate the detailed description of the experimental setup from line 89 to line 117, a restructuring could enhance clarity for the diverse audience of Nature Communications. I suggest opening the section with the overarching objective of the experiments, giving readers a clear context. This could then transition into a relatively detailed introduction of the reference run setup, providing a foundational basis. Following that, a summary of the sensitivity runs, and S1-3 classes could be offered, detailing their main objectives.

Because the full description of the model configuration, including the reference run and sensitivity runs, has been presented in subsection “Model configuration” (L. 495), we think that just the general information and some necessary statements of the experimental design should be presented in the main text. Thus, considering your suggestion, while ensuring clarity and cohesion, we revise the section “Experimental design” as (major revised part is shown here)

<The important role of basal channels in Pine Island ice shelf has been confirmed in distributing basal melting⁴⁷, and in steering and then channeling meltwater out to open waters¹⁵⁻¹⁷. We develop the 3D ISOBC model to investigate channel-influenced ISOBC and basal melting underneath a Pine-Island-like ice shelf, by conducting a suite of idealized numerical experiments (Methods). Nevertheless, the objective of this study is to evaluate the inherent influences brought by an individual basal channel, rather than to reproduce the complicated sub-ice ocean circulation and the associated melting pattern beneath the real Pine Island ice shelf with much more complex basal topography and lateral boundary constraints.

The model uses a uniform vertical resolution of 4 m, which is intermediate between present ice-cavity models and LES⁴⁰. All the simulations are forced by a constant GMW buoyancy influx ($2.3 \text{ m}^4 \text{ s}^{-3}$; Methods) along a stretch of the deepest (southern) boundary, although the basal melting over the whole domain becomes the predominant buoyancy source quickly. In our reference run, for instance, the outflowing channelized buoyancy discharge (Methods) overtakes that GMW buoyancy influx within only 14 h

after the channelized GMW plume first arrives at the northern boundary, and subsequently reaches a quasi-steady value of $296 \text{ m}^4 \text{ s}^{-3}$. The ice shelf base is set to be planar with a prominent basal channel having a constant sinusoidal CSS in the along-slope direction (see, e.g., Fig. 1c, d).

Our idealized numerical experiments consist of one reference run and twenty sensitivity runs in total (Methods). The reference run provides the basic patterns and channelized ISOBC structure, and the sensitivity runs, divided into S1-3 classes, examine the sensitivity of the reference results to a variety of factors, including changing channels CSS, ambient properties, overall basal slope, and vertical resolution (detailed in Table 1 in Methods). In S1, various channel CSSs with different aspect ratios (i.e., the ratio of channel height to width) ...>_L. 113-138.

6. The first paragraph in Discussion (line 354-383) appears to me more fitting for the Introduction, as it outlines the current state of modeling, pointing out the limitations of existing models and positioning the 3D ISOBC model as a novel approach that overcomes many of the limitations of previous approaches.

We incorporate the common content of the first paragraph in the section “Discussion” into the third paragraph in the section “Introduction” as

<Depth-integrated plume models have been an effective numerical tool to investigate the mutual dependence of buoyant plume dynamics and basal channel evolution¹²⁻¹⁴. However, growing evidence from recent high-resolution simulations²⁸⁻³⁰ has demonstrated the importance of capturing the vertical shear and thermohaline structures of subice plumes in order to replicate heat and salt transport from the ambient ocean across the plume to the ice shelf base. Specifically, the turbulent mixing of heat and salt across the ice—ocean boundary layer and the pycnocline separating plume and ambient ocean control the melting-induced cooling and freshening and the entrainment-induced warming and salinization, respectively, of the plumes. These processes depend intrinsically on resolving the details of the vertical stratification and shear^{31,32}. This cannot be achieved in the depth-integrated plume models, in which physical quantities are assumed to be vertically-uniform across the plume and discontinuous at its edges (ref. 33, and derivatives thereof). To overcome this limitation, regional sub-ice cavity models³⁴⁻³⁶ have become an alternative choice to be coupled with ice sheet models. However, these ocean circulation models currently lack the necessary vertical resolution to resolve the vertical structure of ISOBCs, let alone their much coarser representation in circum-Antarctic coupled modelling^{37,38}. While DNS³⁹ (Direct-Numerical-Simulation) and LES (Large-Eddy-Simulation)^{28,29,40-42} ice—ocean boundary layer models may be potential candidates in the future, they have thus far been unable to account for complex large-scale basal topographic features owing to the formidable computational cost associated with their extremely high resolution (in both vertical and horizontal directions).

Consequently, there is a pressing need to develop 3D ice shelf—ocean boundary current (ISOBC) models. This can be a useful compromise with adequate vertical resolution to

capture the key processes occurring at both the upper (ice-ocean) and lower (entrainment) boundaries of the channelized plumes, but computationally efficient enough to be coupled with existing ice sheet models to investigate the intricate interactions between oceanic processes and basal channel evolution. Such an ISOBC model would represent a further development of models from vertical 1D⁴³⁻⁴⁵ to 2.5D (the across-slope gradients of all variables are omitted) vertical slice models⁴⁶ that have been used to investigate the fundamental dynamic and thermodynamic vertical structures of the ISOBC. Those models suggest an upper well-mixed, turbulent layer (the plume region), an intermediate pycnocline suppressing mixing, and an exterior stratified geostrophic flow. However, these low-dimensional ISOBC models cannot investigate the spatiotemporal influence of ice shelf basal channels on the dynamics and thermohaline structure of the ISOBC in an essential three-dimensional way.>_L. 68-105.

7. In several places throughout the manuscript, salinity-related statements such as 'salt discharge' (line 167) and 'salinization' (lines 17 and 401) are mentioned. While I understand that buoyancy is influenced by the density difference arising from variations in salinity and temperature, the manuscript does not provide explicit salinity-related data or figures to support these statements. For the sake of clarity and completeness, it would be beneficial if the authors could: i) Provide explicit data or figures related to salinity, if available; ii) Elaborate on the relationship between the buoyancy figures presented and the salinity-related terms used. This will help readers better grasp the connections and implications.

We add a new figure (Supplementary Fig. 1) of the vertical salinity distribution in the cross section of the northern boundary for the intermediate-width runs, and state that <These changes in the vertical thermal structure are consistent with that in the vertical haline (Supplementary Fig. 1) and thus stratification structures (Supplementary Fig. 2a-d). The density of cold Antarctic waters is conventionally described by a linearized equation of state in which the haline contraction effect dominates over thermal expansion^{33,43-46}. As a result, larger salinity corresponds to denser waters, and thus lower buoyancy, and vice versa.>_L. 286-292.

Supplementary Figure 1. Time-averaged vertical haline structure in the cross section of the northern boundary in **a** H60W8, **b** H140W8, **c** H220W8, and **d** H300W8. The ice base in each plot is marked by the pink line, and the planar part of ice base corresponds to the same ice draft. The default value of 34.36 psu inside the solid ice (i.e., above the planar ice base and outside the channel region) is meaningless. Vertical model grids are also indicated.

8. *The authors provide explicit quantification of channelized basal melting and*

meltwater transport in relation to channel cross-sectional shape (see Figure 3e and Figures 4d, f). While these derived relationships align well with the simulation results from which they originate, their robustness could be further solidified through validation against an independent dataset. For instance, testing against simulations with a range of heights and widths not utilized in the present study would offer additional assurance. Furthermore, discussing potential limitations or conditions of the derived relations would strengthen the study's findings. One specific aspect worth exploring is whether the established relationships remain consistent under varying ambient water conditions.

Thank you for this important point. The derivation of any quantified empirical relations corresponds to certain value ranges of various important parameters. In the present study the basal channel CSS with width from 4 to 12 km and height from 60 to 300 m would be the widest CSS range we can afford based on both the available computational resources and the present model configuration. Therefore, as Reviewer #1 suggested, we should explicitly *reflect the fact that only this range of channel widths and heights were tested (due to resolution limitations)*, the following statements are thus given:

<Consequently, owing to the model resolution limitation, a caveat is not avoidable, as the ranges of channel height and width outside of the present 60 to 300 m and 4 to 12 km, respectively, have not been examined here. ... This quantification demonstrates that both channel height and aspect ratio regulate the overall magnitude of channelized basal melt, the applicability of which, however, needs to be extended for a wider CSS range.>_L. 227-236

and

<Again, an imperative effort needs to be made to extend the applicability of all these empirical expressions of channelized meltwater discharge beyond the present range of CSS.>_L. 369-371.

To this end, the potential limitations or conditions of the derived relations would be, as Reviewer #1 proposed, the relatively rough representation of the shallowest (60 m) and/or narrowest CSSs by using the present $dz=4$ m and $dy=400$ m. Accordingly, we carry out 6 additional runs, that is, H60W8 and H60W12 with $dz=2$ m, H60W4 with both $dz=2$ m and $dy=250$ m, and H140W4, H220W4, and H300W4 with $dy=250$ m, to test drift of the higher resolution simulations w.r.t. their default ones. We list the relevant statistical data in Supplementary Table 1 (see L. 119 in SI), and find that

<For W4 cases excluding H60W4, when the across-slope resolution increases by 60%, the maximal deviations of the channelized melt rate and transport quantities in the higher resolution are only 11% and 21%, respectively. The deviation of H60 cases with wider channels is even less, except for H60W4 that has the largest deviation among all these six runs; the corresponding deviations of the channelized melt rate, discharge, heat discharge, and buoyancy discharge are 22.7%, 30%, 40.7%, and 23.7%, respectively. Consequently, applying the empirical quantitative relations developed here to the basal channels close to the lower limit of cross-sectional area is likely untenable. It is thus worth, as stressed above, exploring the quantified relations between channelized quantities and channel CSS beyond the present CSS range.>_L. 456-466.

In addition, the sensitivity to ambient water properties has been examined in now section “Channelized quantities modulated by ambient properties” (L. 372), although we have to acknowledge that it would be impossible to conduct an exhaustive examination of all the combinations of varying CSS, ambient properties, and basal slope at present.

9. In several places, especially in 'Experimental design', some sentences appear lengthy and densely packed with information. I suggest breaking down particularly complex sentences, such as those discussing the S1 sensitivity runs (lines 103-107) and referencing the findings of Millgate et al. (lines 111-115). This can help improve clarity and ease of reading for the audience.

We revise the sentence

“In S1, various channel CSSs, characterized by different aspect ratios, i.e., H/W , where H and W are channel height and width, respectively, are used to investigate the role of channel geometry in determining channelized melting and GMW channeling capacity, both of which are crucial for ice shelf stability and GMW input into the surrounding ocean.”_l. 103-107

as

<In S1, various channel CSSs with different aspect ratios (i.e., the ratio of channel height to width) are used to investigate the role of channel geometry in channelized melting and GMW channeling capacity. These two quantities are crucial for ice shelf stability and GMW input into the surrounding ocean.>_L. 137-141, and the sentence

“Nevertheless, using a 3D ice-cavity model, Millgate et al.²⁶ found that for narrower channels the geostrophic circulation established within the channel transitions to a slower, ageostrophic overturning circulation, and, accordingly, the simulated basal melt rate is no longer sensitive to the aspect ratio, and is quite limited.”_l. 111-115

as

<Nevertheless, using a 3D ice-cavity model, Millgate et al.²⁶ found that for narrower channels the geostrophic circulation established within the channel transitions to a slower, ageostrophic overturning circulation. The simulated basal melt rate is thus no longer sensitive to the aspect ratio, and is quite limited.>_L. 144-148.

10. The labeling of panels in several figures does not follow a consistent order, typically expected to be from left to right and top to bottom. This inconsistency can reduce clarity and be confusing for readers. For example, in Figure 6, panel b is the first panel on the left in the top row, while panel a is situated as the third panel in that same row. While the authors might have specific reasons for such an unconventional sequence, ensuring clarity for the reader should take precedence.

The layout of panels in Figs. 5 (L. 787), 6 (L. 796) and now Supplementary Figs. 1 (L. 25 in SI), 2 (L. 33 in SI), 8 (L. 115 in SI) is adjusted in a conventional way.

11. In several instances throughout the manuscript, there is an inconsistent use of terms

when referencing locations within figures. Specifically, there is a mix of relative terms like 'left' or 'right' and cardinal directions such as 'eastern' or 'western.' This can be confusing for readers, as the interpretation of 'left' or 'right' depends on the perspective or orientation of the figure. To enhance clarity, I recommend consistently using cardinal directions or other unambiguous terms when referring to locations within figures. This will ensure that readers can quickly and accurately interpret the spatial references made in the text.

We consistently use cardinal directions instead of 'left' or 'right' in the main text, and add the labels of 'west' and 'east' in all the cross-sectional figures.

12. The titles of various sections such as 'Importance of basal channel CSS', 'Modulation by other factors', and 'Impact of vertical resolution' appear somewhat vague. For clarity and a better understanding of the content, I recommend providing more descriptive or specific titles that succinctly capture the main focus or findings of each respective section.

The title of "Importance of basal channel CSS" is revised as <Importance of basal channel cross-sectional shape in ice-ocean interactions>, "Modulation by other factors" is, as Reviewer #1 suggested, divided into <Channelized quantities modulated by ambient properties> and <Channelized quantities modulated by basal slope>, and "Impact of vertical resolution" is revised as <Deviation induced by coarse vertical resolution> (as Reviewer #1 suggested, this section is now Supplementary S1 in L. 57 in SI).

Specific Comments:

Line 22, typo: 'Overturing' should be 'Overturning'.

Revised

Line 55-57, for clarity and accuracy, the correlation should be explicitly stated. It is crucial to provide this distinction, especially when discussing scientific processes where such details matter. In the reference the authors have cited, it is indicated that increased roughness strongly correlates with increased basal melt.

We revise

"In that sense, roughness of ice shelves, that is, basal topographic features spanning a spectrum of wavelengths, strongly correlates with the magnitude of basal melt²⁷."_L. 55-57

as

<In that sense, increased ice shelf basal roughness via topographic features that span a range of length scales, strongly correlates with increased basal melt²⁷.>_L. 62-64.

In lines 57-60, in the concluding sentence, the description 'strength, temperature, and variability of channelized GMW plumes' could benefit from further elaboration. The term 'variability' stands out as particularly broad and ambiguous. It would be beneficial for readers if you could specify the nature of this variability.

“the strength, temperature, and variability of channelized GMW plumes” (l. 59-60) is revised as <the **dynamics and thermohaline structure** of channelized GMW plumes> (L. 66-67).

In line 79-81, the sentence mentioning the separation of the boundary current into an 'exterior geostrophic flow' and an 'interior frictional boundary layer' was derived from a previous study focusing on 2D ice-ocean boundary currents (Jenkins, 2016). The statement suggests that these features are averaged together in depth-integrated plume equations. Its placement in the introduction is unclear, as this study focuses on the 3D extension of the ISOBC model. Can you clarify or better position this statement to align with the main focus of your study?

We revise

“Those models suggest the existence of a near-ice frictional boundary layer and an exterior stratified geostrophic flow, with the former corresponding to an upper weakly stratified layer, i.e., the plume region, and the latter an underlying pycnocline.” l. 78-81

as

<Those models suggest **an upper well-mixed, turbulent layer (the plume region), an intermediate pycnocline suppressing mixing, and an exterior stratified geostrophic flow. However, these low-dimensional ISOBC models cannot investigate the spatiotemporal influence of ice shelf basal channels on the dynamics and thermohaline structure of the ISOBC in an essential three-dimensional way.**> L. 100-105.

In lines 168-174, I recommend substituting 'on one side' and 'on the other side' with terms that better convey the complementary nature of the two sentences.

Considering the specific comment from Reviewer #1 for more precise description of Fig. 3a and b, we revise

“The channelized basal melt rates, area-averaged over the channel domain, are very sensitive to changing channel CSS. On one side, the deepest (300 m) runs exhibit the relatively strongest increasing trend of channelized melt rate with channel width, but the dependence on channel width remarkably diminishes as channel shallows (Fig. 3a). On the other side, the melt rate clearly increases with channel height, but the rate of increase is greater in wider channels (Fig. 3b).” l. 168-174

as

<The channelized basal melt rates, area-averaged over the channel domain, **exhibits strong dependence on** changing channel CSS. **For the shallowest H60 cases, the channelized basal melt is insensitive to channel width, and is the weakest compared**

with other deeper cases (Fig. 3a). For H140 and H220 cases, the channelized basal melt increases with channel width but plateaus beyond W8. For the deepest H300 cases, the channelized basal melt is monotonically enhanced with the channel width. As shown in Fig. 3b, the increasing rate of channelized melt with channel height is larger for wider channels, but this increasing rate becomes less sensitive to channel width from W8 to W12, because H140 and H220 cases plateau beyond W8 (Fig. 3a).>_L. 213-222.

In line 177, rather than stating the intercepts are the 'same', it would be more accurate to describe them as 'similar' or 'close in value', given the numbers 19.46, 20.92, and 19.79 seen in Figure 3c. Additionally, providing an explanation for why these intercepts are closely valued would enhance clarity and depth for the reader.

As we respond above, these similar intercepts arise from the lack of sensitivity to the relatively smallest cross-sectional channel area in H60W4 under the present model resolution; the largest deviations from the higher resolution simulation occur in H60W4. Accordingly, we revise

“Moreover, the regressions of melt rate onto aspect ratio have approximately the same intercept (Fig. 3c).”_l. 176-177

as

<Moreover, the regressions of melt rate onto aspect ratio have similar intercepts (Fig. 3c). In other words, the trend of channelized melt within the present CSS range does not fit the background value of 7.9 m yr⁻¹ in H0 (Fig.3 b, c). Consequently, owing to the model resolution limitation, a caveat is not avoidable, as the ranges of channel height and width outside of the present 60 to 300 m and 4 to 12 km, respectively, have not been examined here.>_L. 225-230.

In line 193, the statement 'In addition, a concomitant near-ice warming also contributes (Fig. 4b, e)' implies that both figures showcase warming. However, it appears that discerning the warming requires a comparison between Fig. 4e and Fig. 4b. To enhance clarity, please consider explicitly stating this or rephrasing the sentence to better illustrate the relationship between the two figures. A similar issue arises with the referencing of Fig. 4a, d, and Fig. 4c, f in line 192. It would be helpful to clarify the relationship between these figures as well.

We revise

“Considering the height-induced difference, the channelized melt is enhanced markedly in extent and magnitude when the channel deepens from 60 to 300 m (Fig. 4a, d), which coincides closely with the difference in friction velocity pattern (Fig. 4c, f). In addition, a concomitant near-ice warming also contributes (Fig. 4b, e).”_l. 190-193

as

<Considering the height-induced difference, the channelized melt is enhanced markedly in extent and magnitude in H300W12 (Fig. 4d), compared with that in H60W12 (Fig. 4a), which coincides with the reinforced friction velocity in H300W12 (Fig. 4f), compared with that in H60W12 (Fig. 4c). In addition, an evidently higher

near-ice temperature in H300W12 (Fig. 4e) than in H60W12 (Fig. 4b) indicates that a concomitant near-ice warming also contributes.>_L. 241-246.

In lines 199-201, the statement highlights the roles of channel height and width in influencing the channelized melt magnitude and extent, respectively. However, it would be beneficial to further elaborate on how channel width constrains the ISOBC dynamics and what is meant by 'the level of constraining' in relation to channel height. Providing more details or a brief explanation can offer clarity for readers less familiar with the ISOBC dynamics.

As may be seen in the amended interpretation of the thermodynamics inside basal channels, a steeper eastern channel flank leads to stronger mixing of GMW with ambient water therein, which results in the reduced buoyancy and thus lower speed of the main channel flow. However, the sidewall slope of channels depends on both channel height and width. So, as suggested in Fig. 3a, the channelized melt rate, representative of the robustness of main channel flow, becomes insensitive to wider channels (8 and 12 km) for runs with channels shallower than 300 m. Accordingly, we state that

<Considering the width-induced difference, the extent of main channelized melt lessens greatly when the channel narrows from 12 to 4 km (Fig. 4g), which is led by the reduced friction velocity (Fig. 4i). *In detail, as demonstrated in the next section, a steeper eastern channel flank induces stronger mixing of GMW with ambient water therein, which results in the reduced buoyancy and thus lower speed of the main channel flow. ... This suggests that channel height determines the channelized melt magnitude by controlling both determinants, and channel width influences the channelized melt extent by constraining the ISOBC dynamics. Because the sidewall slope of channels depends on both channel height and width, the channelized melt rate, representative of the robustness of main channel flow, becomes insensitive to wider channel (8 and 12 km) for runs with channel shallower than 300 m (Fig. 3a).*>_L. 246-259.

In lines 202-206: While the subsequent description of the evolution of discharge-averaged heat provides some insight into why the channelized outflowing quantities might be of interest, the statement in line 203 about these quantities being of 'particular concern for oceanographers and climatologists' would benefit from a direct link or reference to the overarching implications or concerns of these fields. This strengthens the narrative and ensures continuity for the reader.

We revise

“Besides the channelized basal melting, we concentrate on the channelized outflowing quantities that are of particular concern for oceanographers and climatologists.”_1. 202-204

as

<Besides the channelized basal melting, we concentrate on the channelized outflowing quantities that are of particular concern *for assessing the influence of ice shelf basal*

meltwater on near-Antarctic waters^{50,51}, Southern Ocean^{52,53}, and the global climate system⁵⁴⁻⁵⁶.>_L. 260-263.

In line 214, the description 'flat and sharp' regarding the lower interface of the thermocline is somewhat ambiguous. Can you provide additional context or clarification on what is meant by this combination of descriptors? Specifically, how can the interface simultaneously be both flat and sharp?

We revise

“The upper interface of the underlying thermocline is curved caused by the channel geometry, but its lower interface is relatively much flat and sharp.”_L. 213-214

as

<The upper interface of the underlying thermocline is curved **into the channel**, caused by the channel geometry, but its lower interface is, **in contrast, flatter and vertically sharper**.>_L. 276-279.

In line 262, the term 'accurately' may give the impression of a perfect or near-perfect fit. However, upon examining Fig. 7a, I observed some discrepancies between the simulated data points and the fitted curve. I would suggest using more appropriate words to describe the fit, such as 'reasonably well' or 'with a good degree of approximation,' to more accurately represent the observed relationship.

We revise “is quantified accurately enough with respect to” as <is quantified **reasonably well** with respect to>.

In lines 284-286, the authors identify the 'ice-shelf basal slope and ambient water properties' as important factors influencing channelized melt and transport. For clarity and to provide context to readers, I recommend offering a brief rationale for the significance of these factors, supported by relevant literature or previous findings.

We revise

“The purpose of designing the S2 experiments is to investigate and quantify the influences of two important factors, ice-shelf basal slope and ambient water properties, on the channelized melt and transport.”_L. 284-286

as

<The purpose of designing the **S2-A** experiments is to investigate and quantify the influences of ambient water properties on the channelized melt and transport, **in view of the crucial entrainment process from the ambient water**^{30,44,46}.>_L. 373-375

and

<**Besides the channel CSS, ice-shelf basal slope is another topographic determinant of buoyant meltwater plume forcing**⁴³⁻⁴⁶, so the sensitivity of channelized melt and transport to basal slope is examined here.>_L. 392-394.

In lines 333-334, the authors mention 'The foremost reason for the artificially enhanced

mixing is the exaggerated vertical diffusion...'? A more detailed explanation for this assertion is needed. While coarse vertical resolution can introduce biases in vertical mixing, the direction and magnitude of the bias (whether it enhances or reduces mixing) depend on various factors, such as the specifics of the turbulence closure scheme and the physical processes being represented. In addition, a clarification why the vertical diffusion is more pronounced on the eastern channel flank in the S1 runs compared to the S3 runs.

In lines 336-337, it is hard to see from Figure 8 that 'As the channel deepens, the progressive reduction in the vertical diffusion in the S3 runs', due to the color bar's upper limit. Providing quantitative metrics or adjusting the visualization might offer a clearer understanding of this observation.

Thank you for these important points. Based on the amended interpretation of the thermodynamics inside the channels, the discussion of the deviation induced by coarse vertical resolution is substantially revised (see the following excerpt). In general, using the inferior vertical resolution cannot resolve the near-ice Ekman layer east of the channel, poorly resolves the pycnocline structure inside the channel, and overly relieves the stratification adjacent to the western channel base. These deficiencies respectively lead to the low vertical diffusivity east of the channel, attenuated secondary overturning, and overestimated vertical diffusivity along the western channel flank. Nevertheless, all these deviations are reduced for deeper (i.e., 220 and 300 m here) basal channels.

<First, the most robust GMW plume and its underlying pycnocline structure in the shallowest channel (Supplementary Fig. 2a) cannot be reproduced by using the inferior vertical resolution (Supplementary Fig. 2e), which is responsible for the largest thermohaline differences indicated in Supplementary Fig. 7d and e. In detail, in H60W8-C, except for the main channel flow area, the near-ice Ekman layer of GMW cannot be resolved at all (Supplementary Fig. 2e), which corresponds to the low vertical diffusivity therein (Supplementary Fig. 8a). In addition, the upper and lower interfaces of the underlying pycnocline with pronounced vertical gradients also cannot be reproduced, instead merging into an over-relaxed single pycnocline (Supplementary Fig. 2e) that corresponds to the exaggerated vertical diffusion (Supplementary Fig. 8a), compared with that in H60W8 (Fig. 6a). When the channel is deepened to 140 m, i.e., in H140W8-C, the near-ice Ekman layer still cannot be resolved for the regions outside the main channel flow (Supplementary Figs. 2f and 8b). The upper interface of the pycnocline can be formed (Supplementary Fig. 2f) but is much less tilted than that in H140W8 (Supplementary Fig. 2b). The topographic secondary overturning emerges but is fairly attenuated (Supplementary Fig. 8f), compared with that in H140W8 (Fig. 6f), because the steepness of the isopycnals is much relieved with the larger vertical spacing (Supplementary Fig. 2b, f). Moreover, the prominent stratification along the recirculation of iGMW (Supplementary Fig. 2b) is markedly relieved in H140W8-C (Supplementary Fig. 2f), which results in the corresponding overestimated vertical diffusivity along the western channel flank (Supplementary Fig. 8b). When the channel is deepened further, the stratification structure (Supplementary Fig. 2g, h) and dynamic processes (Supplementary Fig. 8c, d, g, h) for the coarser vertical resolution

progressively resemble that for the higher vertical resolution (Supplementary Fig. 2c, d and Fig. 6c, d, g, h), including the occurrence of a poorly resolved near-ice Ekman layer of the lateral GMW inflow from the east of channel. The shown comparability thus explains the aforementioned decrease of the differences in the discharge-averaged heat and buoyancy.>_L. 76-104 in SI.

Thus, as you review above, the statement in l. 336-339 has been removed.

In lines 347-352, the transition from the analysis of the S3 runs to the broader critique of 2D plume models is abrupt and lacks a clear connecting rationale.

As suggested, the discussion of 2D plume models is now moved to the first section “Introduction”.

In lines 453-455, it should be mentioned that this setup is for the reference run.

Revised as suggested.

Figures

In several across-slope figures, such as Fig. 2 (panels a, d, and g) and Fig. 6 (panels c and d), there’s a noticeable staircase-like pattern in the shading adjacent to the pink line. Can the authors provide clarification on the origin of this pattern?

We add a statement in the caption of Fig. 2 as

<...Vertical model grids are also indicated. The staircase-like patterns adjacent to the channel base are the inherent artifact of using the structured z grid in our ISOBC model.>_L. 769-771.

Response to Reviewer #3:

Overall comments:

The article describes the results from a new 3D ice shelf-ocean boundary current model, which represents vertical mixing in meltwater plumes beneath the ice shelf. The work is important due to the role of plumes and channels in ice shelf stability, and the work has the potential to aid our understanding of the processes that are going on. The findings over the influence of “cross-sectional shapes” on melt rates were noteworthy. The work has the potential to be of significance to both glaciology and oceanography communities.

We thank the reviewer for their support.

So, in principle the work is worth publication, however, in the format it is in presently, the manuscript doesn't show off the work to the best of its ability. There is a lot of qualitative/descriptive words in the results sections, describing all the results in detail rather than drawing the reader to the important points.

Thank you this important comment, and we try to elucidate the important points properly after the corresponding qualitative or descriptive parts. Here are the most significant instances:

- After describing the along-slope evolution of vertical structures of buoyancy and velocity shown in Fig. 2, that is, <In detail, an ISOBC core leaning on the Coriolis-favored side of the channel gradually emerges, with increasingly larger buoyancy (Fig. 2a, d, g), more downward tilted isopycnals (Fig. 2a, d, g), and faster along-slope velocity (Fig. 2b, e, h).> (L. 192-195), we conclude that <Accordingly, the geostrophy of along-slope velocity inside the channel is verified here: an increase in the across-slope pressure gradient corresponding to the gradually tilted isopycnals beneath the western part of channel accompanies the collocated stronger along-slope velocity.> (L. 195-199). That is the prerequisite for our later analysis of the thermodynamics inside basal channels.
- Following the observation that <the regressions of melt rate onto aspect ratio have similar intercepts (Fig. 3c). In other words, the trend of channelized melt within the present CSS range does not fit the background value of 7.9 m yr^{-1} in H0 (Fig.3 b, c).> (L. 225-227), we propose a necessary caveat statement that <owing to the model resolution limitation, a caveat is not avoidable, as the ranges of channel height and width outside of the present 60 to 300 m and 4 to 12 km, respectively, have not been examined here.> (L. 227-230). In addition, we conclude that <This quantification demonstrates that both channel height and aspect ratio regulate the overall magnitude of channelized basal melt, the applicability of which, however, needs to be extended for a wider CSS range.> (L. 234-236).
- After we describe the changes in the basal melt pattern caused by varying channel width, that is, <the extent of main channelized melt lessens greatly when the channel narrows from 12 to 4 km (Fig. 4g), which is led by the reduced friction velocity (Fig.

4i).> (L. 247-249), we now expound that  (L. 249-251).

- After a revised elaborate description of changes in the vertical thermal structure with deepening channel shown in Fig. 5 (L. 272-286), we argue that <All these thermohaline changes are closely tied to the hydrodynamic changes within the channels as they are deepened.> (L. 293-294), immediately followed by our amended interpretation of the thermodynamics inside basal channel (L. 294-336).

There were a lot of “appears to” statements, which needed to be more definite statements, and “relatively”s that needed quantifying.

Agree. We revise

“Simultaneously, the ISOBC’s overflow across the channel western boundary appears to be greater towards the calving ice front.”_l. 141-142

as

<Simultaneously, the GMW flowing across the channel western boundary becomes greater towards the calving ice front.>_L. 177-178,

“The predominant dependence of discharge-averaged channelized outflow heat on basal channel height (Fig. 5a) appears to be described by a log-law for the intermediate-width runs (blue component in Fig. 7a).”_l. 253-255

as

<That dependence for the intermediate-width runs can be simply described by a log-law, allowing for both a concise form and its more desirable performance than a power law (blue component in Fig. 8a).>_L. 339-341,

“As shown in Fig. 7c, even though the dependence of the normalized discharge p.u.w. on channel aspect ratio appears to be sensitive to channel width, ...”_l. 264-266

as

<As shown in Fig. 8c, even though the dependence of the normalized discharge p.u.w. on channel aspect ratio is sensitive to channel width, ...>_L. 352-353,

“..., while the difference in the friction velocity pattern appears to be less noticeable (Supplementary Fig. 5-C5, 6).”_l. 319-321

as

<..., while the difference in the friction velocity pattern is less noticeable (Supplementary Fig. 6-C5, 6).>_L. 64-65 in SI, and

“Overall, the adoption of coarser vertical resolution appears to artificially generate more intense mixing inside the channel.”_l. 328-329

as

<Overall, the adoption of coarser vertical resolution artificially generates warmer and saltier water masses inside the channel.>_L. 71-73 in SI.

In addition, we revise

“..., we find that a fully-developed relatively coldest (and definitely freshest) mixed layer builds up ...”_l. 210-211

as

<..., we find that a fully-developed relatively cold (lower than $-1.4\text{ }^{\circ}\text{C}$) mixed layer builds up ...>_L. 274-275,

“The upper interface of the underlying thermocline is curved caused by the channel geometry, but its lower interface is relatively much flat and sharp.”_L. 213-214

as

<The upper interface of the underlying thermocline is curved into the channel, caused by the channel geometry, but its lower interface is, in contrast, flatter and vertically sharper.>_L. 276-279, and

“the channelized discharge p.u.w. and heat discharge p.u.w. can be readily quantified in terms of channel height and aspect ratio, which suggests that deep basal channels with steep side slopes can lead to both relatively large mass and heat discharge p.u.w.”_L. 269-271

as

<the channelized discharge p.u.w. and heat discharge p.u.w. can be readily quantified in terms of channel height and aspect ratio, which suggests that deep basal channels with steep side slopes can lead to large mass and heat discharge p.u.w, compared with shallow channels with more gradual side slopes.>_L. 355-359.

At last, we make quantification for new occurrences of the “relatively”s if necessary.

Some features of the figures were described in depth, an indication of these particular areas on the figures would be useful.

As you may review our amended thermodynamic analysis (L. 293-336) responsible for the changes in the vertical thermohaline structure inside the deepening channel (L. 272-292), we summarize and highlight the associated critical features in the following schematic figure (i.e., Fig. 7).

Figure 7. Schematic comparison of stratification and hydrodynamics within shallow and deep channels. Shallow channels (left) have little influence on the

properties of the across-channel GMW flow. Little GMW is recirculated upwards along the western channel flank. In contrast, the stratification and hydrodynamics are significantly changed in deep channels (right) by four critical coherent processes as illustrated. The arrow colour transitioning from blue to pinkish red denotes the transformation of GMW to warmer iGMW, and vice versa. Not to scale.

I was hopeful the discussion would draw the results together, and while there were some summary statements, there were also some areas of vagueness. For example, lines 392-396 make a big statement about profound impacts, but doesn't elaborate what the impacts are. Therefore, it is not so much that the work doesn't "support" the conclusions and claims, but it is that the conclusions and claims could be better connected to the results.

Thank you for this important comment that we totally agree with. Therefore, we thoroughly rewrite the section "Discussion" to improve the connection of our results to the relevant implications, including some revisions made for the specific concerns of Reviewers #1 and #2:

<Channelized basal melting of ice shelf is shown here to be substantially controlled by the channel CSS, which can be quantified in terms of channel height and aspect ratio (Fig. 3e). Therefore, the necessity of utilizing historical remote sensing observations of basal channel geometry and contemporary channelized melting^{8,20} to verify this empirical quantification is emphasized. That provides a possibility of inferring an overall channelized melt rate derived from the oceanic perspectives (i.e., the 3D ISOBC model), rather than the remote sensing-based methods, for the current Antarctic ice shelf basal topography.

Nevertheless, investigating the detailed evolution of basal channel system influenced by changing oceanic conditions relies crucially on the ice sheet-ocean coupled models, rather than the stand-alone ice sheet models (e.g., refs. 58,59). Within these stand-alone ice sheet models, an idealized, symmetric melt rate distribution across the channel is normally prescribed by using a Gaussian function centered at the channel apex. Such a symmetric pattern deviates substantially from the asymmetric pattern demonstrated here and in observations (e.g., refs. 8,20). As claimed in Wearing et al.⁵⁹, "further complexity may arise through dependence on depth, basal slope, along-flow position, ocean heat content, plume entrainment rates, Coriolis effects and ice-shelf cavity circulation". To this end, our 3D ISOBC model, as previously mentioned, may be one desirable candidate for the ocean component to be coupled with existing ice sheet models. To further demonstrate the need to capture the vertical structure of the channelized ISOBC in such a model, the influence of adopting a coarser (10 m vs 4 m) vertical resolution is examined in S3 (see Supplementary S1). It is found that the deficiency of vertical resolution results in poorly resolving the near-ice turbulence and meltwater-induced stratification, as well as the pycnocline structure inside shallower (60 and 120 m) channels, which jointly influence the assessment of channelized quantities.

In addition, the outflowing meltwater from beneath the Antarctic ice shelves has become a topic of great attention in recent years because of its important role, as sketched in Fig. 9, in coastal polynya formation^{8,15-17}, shelf water modification^{18,19},

shelf circulation variability^{60,61}, and sea ice growth/melting^{62,63}, and in turn even has profound impacts on the global climate system⁵⁴⁻⁵⁶. The evaluation of all these influences exerted by the Antarctic basal meltwater is generally implemented by using a variety of global coupled models in which, however, there are great uncertainties in the location and amounts of applied basal meltwater influx⁵⁰⁻⁵⁶. The prominent basal channels propagating to the ice shelf calving front are important conduits of basal meltwater to the open seas. Therefore, the quantification of channelized outflowing mass, heat, and buoyancy discharge of meltwater established here (Fig. 6c, d, f) has the potential to reduce these uncertainties.

However, considering the relatively poor representation of these channel CSS using the current resolution ($dy=400$ m, $dz=4$ m), it is worthwhile to examine the reliability of quantified relations between channelized quantities and channel CSS proposed here by performing higher resolution ($dy=250$ m, $dz=2$ m) simulations for the shallowest H60 and the narrowest W4 cases. Thereupon, six additional runs including H60W8 and H60W12 with $dz=2$ m, H60W4 with both $dz=2$ m and $dy=250$ m, and H140W4, H220W4, and H300W4 with $dy=250$ m are conducted, and the statistics of the simulated channelized quantities are summarized in Supplementary Table 1. For W4 cases excluding H60W4, when the across-slope resolution increases by 60%, the maximal deviations of the channelized melt rate and transport quantities in the higher resolution are only 11% and 21%, respectively. The deviation of H60 cases with wider channels is even less, except for H60W4 that has the largest deviation among all these six runs; the corresponding deviations of the channelized melt rate, discharge, heat discharge, and buoyancy discharge are 22.7%, 30%, 40.7%, and 23.7%, respectively. Consequently, applying the empirical quantitative relations developed here to the basal channels close to the lower limit of cross-sectional area is likely untenable. It is thus worth, as stressed above, exploring the quantified relations between channelized quantities and channel CSS beyond the present CSS range. Furthermore, although these relations are derived under idealized configurations that undoubtedly differ from the real circumstances including a wide range of basal conditions and ice shelf dynamics, they may provide preliminary estimates with credible magnitude.

This study demonstrates that deep channels may correspond to substantial amplification in channelized basal melting, meltwater channeling, and plume warming and salinization (as illustrated in Fig. 9). In other words, there might exist a transition into warmer and saltier near-base hydrography if basal channels deepen, which is attributed to the unique changes in the thermodynamic processes and concomitant thermohaline structure inside the deepening channels (Fig. 7). Therefore, there may be a pressing need for the comprehensive hydrographic and turbulence-related surveys in the ice shelf basal channels, in addition to the improvement in observing the evolving basal channels and the hydrography adjacent to the calving ice front where channelized meltwater outflows.>_L. 410-479.

I think in summary, the paper has promise, the model looks great with a sound methodology, and there's some interesting results, but the paper needs a tighter focus. I'm not sure if it is a model description paper (that's what the results section feels like) or providing a clear result with important implications (what the discussion feels like it is trying to do). I was not much wiser at the end of the discussion about the advances made to the big picture topics outlined in the introduction. I think with a bit of a re-write to be more focused, this could be a neat paper.

In the first section “Introduction” of this revised manuscript, we first elucidate the importance of basal channels for the overall ice shelf stability and the interactions between the channel and channelized meltwater plume. Then we compare in detail the plume, sub-ice cavity, and 3D ISOBC models, and draw a conclusion that the 3D ISOBC model developed here can be a useful tool to study the interactions between oceanic processes and basal channel evolution. In the second section “Results”, in which the influences of basal channel on basal melting and ISBOC are systematically studied, we substantially revise the description, analysis and discussion of the simulated results for further clarity and accuracy. Specifically, we deliberately amend the thermodynamic analysis in the cross section of channel. At last, as we respond to your previous comment, the connection of our results to the relevant implications expanded upon in the last section “Discussion” is hopefully enhanced. In summary, we hope all the revisions made in this revised manuscript make sense, after carefully considering each comment of every reviewer.

Figures:

The contour labels on plan figures (e.g. Fig.1) are impossible to read – these need to be much bigger

The contour labels in Figs. 1 (L. 757), 4 (L. 782) and Supplementary Figs. 3 (L. 40 in SI) and 6 (L. 106 in SI) are enlarged.

Why are g-i) smaller in Figure 4? Was the overall domain narrower as well as the channel width?

Yes. Although we have stated that “In the across-slope (y) direction, the extent of the planar base part adjacent to each side of the channel area is set to 10 km, so the total across-slope extent is 20 km plus the specific channel width.” in the subsection “Model configuration” (l. 423-426), we add a statement <Note that the extent of the planar base part adjacent to each side of the channel area is consistently set to 10 km for all runs here.> in the caption of Fig. 4 (L. 784-785).

The numbering system on Figs 5 & 6, 8 is a bit muddled and hard to follow.

The layout of panels in Figs. 5 (L. 787), 6 (L. 796) and now Supplementary Fig. 8 (L. 115 in SI) is adjusted in a conventional way.

Original references selected

7. Alley, K. E., Scambos, T. A. & Alley, R. B. The role of channelized basal melt in ice-shelf stability: recent progress and future priorities. *Ann. Glaciol.* 1-5 (2023).
8. Alley, K. E., Scambos, T. A., Siegfried, M. R. & Fricker, H. A. Impacts of warm water on Antarctic ice shelf stability through basal channel formation. *Nat. Geosci.* **9**, 290-293 (2016).
16. Mankoff, K. D., Jacobfs, S. S., Tulaczyk, S. M. & Stammerjohn, S. E. The role of Pine Island Glacier ice shelf basal channels in deep-water upwelling, polynyas and ocean circulation in Pine Island Bay, Antarctica. *Ann. Glaciol.* **53**, 123-128 (2012).
43. Jenkins, A. A simple model of the ice shelf—ocean boundary layer and current. *J. Phys. Oceanogr.* **46**, 1785-1803 (2016).
54. Golledge, N. R. et al. Global environmental consequences of twenty-first-century ice-sheet melt. *Nature* **566**, 65-72 (2019).
55. Li, Q., England, M. H., Hogg, A. M., Rintoul, S. R. & Morrison, A. K. Abyssal ocean overturning slowdown and warming driven by Antarctic meltwater. *Nature* **615**, 841-847 (2023).
41. in 1. 600 Wåhlin, A. K. Downward channeling of dense water in topographic corrugations. *Deep Sea Res. Part I Oceanogr. Res. Pap.* **51**, 577-590 (2004).

Additional references

47. Shean, D. E., Joughin, I. R., Dutrieux, P., Smith, B. E. & Berthier, E. Ice shelf basal melt rates from a high-resolution digital elevation model (DEM) record for Pine Island Glacier, Antarctica. *Cryosphere* **13**, 2633-2656 (2019).
50. Moorman, R., Morrison, A. K., & Hogg, A. M. Thermal responses to Antarctic ice shelf melt in an eddy-rich global ocean–sea ice model. *J. Clim.* **33**, 6599-6620 (2020).
51. Thomas, M. et al. Future response of Antarctic Continental Shelf temperatures to ice shelf basal melting and calving. *Geophys. Res. Lett.* **50**, e2022GL102101 (2023).
52. Rye, C. D. et al. Antarctic glacial melt as a driver of recent Southern Ocean climate trends. *Geophys. Res. Lett.* **47**, e2019GL086892 (2020).
53. Gorte, T., Lovenduski, N. S., Nisssen, C., & Lenaerts, J. T. M. Antarctic Ice Sheet freshwater discharge drives substantial Southern Ocean changes over the 21st century. *Geophys. Res. Lett.* **50**, e2023GL104949 (2023).
56. Bronselaer, B. et al. Change in future climate due to Antarctic meltwater. *Nature* **564**, 53-58 (2018).

REVIEWER COMMENTS

Reviewer #1 (Remarks to the Author):

The authors have made considerable improvements including additional numerical model simulations, a schematic, and numerous improvements to the discussion and existing figures. Therefore, I believe the manuscript is ready for publication after revisiting the subsequent follow-up comment.

Regarding the discussion on western-eastern re-entrant/periodic boundary conditions (L509-520), these new updates do clarify the choice of re-entrant boundary conditions. However, one point is still lacking in clarity - based on your summary of the simulation (not shown) using closed east/west boundaries, you make the case that the existence of closed boundaries and their proximity of the channel to the boundary are in fact, important parameters to consider and are part of the real ice-shelf meltwater conduit system that isn't represented/tackled in this study (this point just doesn't yet seem clear in the current discussion). Perhaps you agree with the following statement: To fully appreciate the dynamics of the sub-ice shelf freshwater transport, the theories/model results in this work (in the periodic, channel-isolated framework) should also incorporate finite (east-west) channel width and channel-boundary distance as additional test parameters (perhaps in a future study)?

Minor suggestions: L509: Delete "Actually"

L514: "the decreasing distance" -> "decreasing distance"

Reviewer #1 (Remarks on code availability):

The code availability and documentation is adequate for reproducibility of the results provided in the manuscript.

Reviewer #2 (Remarks to the Author):

Please see the attachment.

Reviewer #3 (Remarks to the Author):

In my previous review, my main comments related to the presentation of the paper and some elements of the figure presentation. Looking over the rebuttal & resubmitted manuscript, the authors have addressed these concerns suitably.

One comment on structure however: the authors have added a section to the discussion relating to model resolution - this is introducing new results, so seems a bit disjointed sitting in the discussion section. I suggest a short section introducing these results is added to the results section, rather than introducing them in the discussion section. The results can then be discussed without disrupting the flow of the discussion.

Response to reviewers:

Ice shelf basal channel shape determines channelized ice-ocean interactions

Chen Cheng, Adrian Jenkins, Paul R. Holland, Zhaomin Wang, Jihai Dong, Chengyan Liu

We are grateful to Qin Zhou and other two anonymous reviewers for their very constructive and helpful comments and suggestions in the 2nd round of review. In the following, *italic* denotes the reviewers' comments, and the following is our response. "l. #" and "L. #" denote line # in the previous and revised manuscripts, respectively. Double quote and angle bracket represent the excerpts of the previous and revised manuscripts, respectively. All the changes from the previous manuscript are shown in blue characters in the attached file 'manu2-r2-showing changes.pdf'.

Response to Reviewer #1:

The authors have made considerable improvements including additional numerical model simulations, a schematic, and numerous improvements to the discussion and existing figures. Therefore, I believe the manuscript is ready for publication after revisiting the subsequent follow-up comment.

We thank the reviewer for their support.

Regarding the discussion on western-eastern re-entrant/periodic boundary conditions (L509-520), these new updates do clarify the choice of re-entrant boundary conditions. However, one point is still lacking in clarity - based on your summary of the simulation (not shown) using closed east/west boundaries, you make the case that the existence of closed boundaries and their proximity of the channel to the boundary are in fact, important parameters to consider and are part of the real ice-shelf meltwater conduit system that isn't represented/tackled in this study (this point just doesn't yet seem clear in the current discussion). Perhaps you agree with the following statement: To fully appreciate the dynamics of the sub-ice shelf freshwater transport, the theories/model results in this work (in the periodic, channel-isolated framework) should also incorporate finite (east-west) channel width and channel-boundary distance as additional test parameters (perhaps in a future study)?

Thank you for this important comment that we totally agree with. We revise
“The northern boundary is set as an open boundary, with zero-gradients, while the western and eastern boundaries are reentrant, i.e., the periodic lateral boundaries. Actually, the closed western and eastern boundaries have been tested, and the former has more significant effects on the simulated results (not shown). The meltwater piles up against the western boundary, and outflows along this boundary as a “western boundary current”. Thus, the western boundary—ice front corner becomes the other main exit of meltwater. Moreover, the piled meltwater can also extend into the channel with the decreasing distance from the western boundary to the channel. To this end, setting the western and eastern boundaries to be periodic allows us to avoid these undesired uncertainties that significantly obscure the channeling function. In addition, the periodic lateral boundaries enable us to form a consistent, permanent, and well-developed meltwater-cycling system within our domain, so the role the basal channel plays in the ice-ocean interactions can be evaluated adequately.”_1. 507-520

as

<The northern boundary is set as an open boundary, with zero-gradients, while the western and eastern boundaries are reentrant, i.e., the periodic lateral boundaries that enable us to form a consistent, permanent, and well-developed meltwater-cycling system within our domain. Thus, the role the basal channel plays in the ice-ocean interactions can be evaluated adequately in the channel-isolated framework. Nevertheless, the closed western and eastern boundaries have also been tested, and the former has more significant effects on the simulated results (not shown). The meltwater

piles up against the western boundary, and outflows along this boundary as a “western boundary current”. *As such*, the western boundary—ice front corner becomes the other main exit of meltwater. Moreover, the piled meltwater can also extend into the channel with decreasing distance from the western boundary to the channel. *The existence of closed boundaries and the proximity of the channel to them are in fact the part of some real basal meltwater conduit systems, which therefore should be incorporated in a future real-case study to supplement the present findings.*>_L. 499-513.

Minor suggestions: L509: Delete "Actually"

Revised as you see above.

L514: "the decreasing distance" -> "decreasing distance"

Revised as you see above.

Response to Reviewer #2 (Qin Zhou):

General Comments:

I appreciate the great efforts made by the authors in addressing the concerns and suggestions raised in the first round of reviews. The revisions have notably improved the manuscript. However, there are still a few areas that require further attention to ensure the manuscript meets the publication's standards. Once these remaining issues are adequately addressed, I would support the acceptance of the manuscript for publication.

We thank the reviewer for their support.

Comments:

In lines 82-90, where the authors address limitations of the regional cavity models, the discussion would benefit from a closer examination within the context of this study, perhaps by focusing on detailing the thermal structure associated with the plume.

Thank you for this important comment that we totally agree with. We revise “However, these ocean circulation models currently lack the necessary vertical resolution to resolve the vertical structure of ISOBCs, let alone their much coarser representation in circum-Antarctic coupled modelling^{37,38}. While DNS³⁹ (Direct-Numerical-Simulation) and LES (Large-Eddy-Simulation)^{28,29,40-42} ice—ocean boundary layer models may be potential candidates in the future, they have thus far been unable to account for ...”_l. 82-87

as

<However, these ocean circulation models currently lack the necessary vertical resolution to resolve the vertical structure of **ice shelf—ocean boundary currents (ISOBCs), i.e., the upper relatively well-mixed plume region and its underlying pycnocline transitioning to the ambient water,** let alone their much coarser representation³⁷ **or complete neglect**³⁸ in circum-Antarctic coupled modelling. While DNS³⁹ (Direct-Numerical-Simulation) and LES (Large-Eddy-Simulation)^{28,29,40-42} ice—ocean boundary layer models may be potential candidates in the future, **which can adequately resolve the ISOBC structure,** they have thus far been unable to account for ...>_L. 82-90

In line 84, where 'ISOBC' is mentioned for the first time, it should be presented with its full name for clarity.

Revised as you see above.

In line 133, the term 'basic patterns' is used ambiguously. Could you specify the patterns being referred to?

We revise

“The reference run provides the basic patterns and channelized ISOBC structure, ...”_l. 133-134

as

<The reference run provides the basic **basal melting and upper plume** patterns and channelized ISOBC structure, ...>_L. 134-135.

In lines 215-222, clarity in describing the results could be enhanced by specifying 'widens from 8 km to 12 km' instead of the vague term 'W8 to W12'. Additionally, I recommend that the authors thoroughly review the manuscript to rectify similar instances of potential ambiguity.

Sorry for that ambiguity. We revise

“For the shallowest H60 cases, the channelized basal melt is insensitive to channel width, and is the weakest compared to other deeper cases (Fig. 3a). For H140 and H220 cases, the channelized basal melt increases with channel width but plateaus beyond W8. For the deepest H300 cases, the channelized basal melt is monotonically enhanced with the channel width. As shown in Fig. 3b, the increasing rate of channelized melt with channel height is larger for wider channels, but this increasing rate becomes less sensitive to channel width from W8 to W12, because H140 and H220 cases plateau beyond W8 (Fig. 3a).”_l. 215-222

as

<For the shallowest (**60 m**) cases, the channelized basal melt is insensitive to channel width, and is the weakest compared to other deeper cases (Fig. 3a). For **the intermediate height (140 and 220 m)** cases, the channelized basal melt increases with channel width but plateaus **when channel widens** beyond 8 km. For the deepest (**300 m**) cases, the channelized basal melt is monotonically enhanced with the channel width. As shown in Fig. 3b, the increasing rate of channelized melt with channel height is larger for wider channels, but this increasing rate becomes less sensitive to **the increasing** channel width from **8 to 12 km**, because **the intermediate height** cases, **as mentioned above, level off for wider channel cases** (Fig. 3a).>_L. 216-224.

After thoroughly review the manuscript, we additionally revise

“... simulations for the shallowest H60 and the narrowest W4 cases. Thereupon, six additional runs including ...”_l. 451-452,

“For W4 cases excluding H60W4, when the across-slope resolution increases by 60%, ...”_l. 456-457, and

“The deviation of H60 cases with wider channels is even less, except for H60W4 that ...”_l. 458-459

as

<... simulations for the shallowest (**60 m**) and the narrowest (**4 km**) cases. Thereupon, six additional runs including ...>_L. 62-63 in SI,

<For **the narrowest** cases excluding H60W4, when the across-slope resolution increases by 60%, ...>_L. 66-67 in SI, and

<The deviation of the shallowest cases with wider channels is even less, except for H60W4 that ...>_L. 69-70 in SI, respectively.

In lines 310-312, the authors state an eastward Ekman transport that isn't immediately apparent in Figures 2i and 6f. To support this statement, a more detailed explanation is necessary to connect what is shown in the figures with the textual statement.

In the previous revised manuscript, we have given the corresponding detailed explanation immediately following that statement in l. 310-312. Nevertheless, as you commented, their connection is vague. So, we rearrange the elaboration

“Although most of the iGMW crosses the channel (Fig. 2f, i), some iGMW is recirculated by a weak eastward Ekman transport adjacent to the western channel flank (Figs. 2i, 6f). Since the geostrophy of along-slope velocity inside the channel has been verified in Fig. 2, the occurrence of this recirculation can be regarded as a dynamic response to a larger across-channel pressure gradient induced by more tilted isopycnals and steeper topography (Supplementary Fig. 2b) than that in the shallow (60 m) channel (Supplementary Fig. 2a). In other words, larger across-channel pressure gradient results in larger geostrophic along-channel flow that leads to an increasingly significant near-ice Ekman transport directed to the east (Southern Hemisphere) of this main channel flow. The recirculation of iGMW upward along the western channel flank makes the water, interacting with the ice base, warmer (Fig. 5c) and saltier (Supplementary Fig. 1b) than that in H60W8 (Fig. 5b and Supplementary Fig. 1a). This recirculation, combined with ...”_l. 310-322

as

<Since the geostrophy of along-slope velocity inside the channel has been verified in Fig. 2, an eastward Ekman transport emerges adjacent to the western channel flank (Figs. 2i, 6f). In detail, this feature can be regarded as a dynamic response to a larger across-channel pressure gradient induced by more tilted isopycnals and steeper topography (Supplementary Fig. 2b) than that in the shallow (60 m) channel (Supplementary Fig. 2a). In other words, larger across-channel pressure gradient results in larger geostrophic along-channel flow that leads to an increasingly significant near-ice Ekman transport directed to the east (Southern Hemisphere) of the main channel flow. Therefore, most of the iGMW can cross the channel (Fig. 2f, i), some iGMW is, however, recirculated by the eastward Ekman transport, which makes the water, interacting with the western-channel ice base, warmer (Fig. 5c) and saltier (Supplementary Fig. 1b) than that in H60W8 (Fig. 5b and Supplementary Fig. 1a). This recirculation, combined with ...>_L. 313-325.

In lines 289-292, the discussion on vertical haline and buoyancy structures, which addresses my previous general comment No.7, would be more contextually appropriate in the sections dealing with 'salt discharge' or 'salinization'.

The term 'salt discharge' only appears once at the very beginning of the primary subsection “Importance of basal channel cross-sectional shape in ice-ocean

interactions”, that is, “The purpose of the S1 experiments is to ... on the channelized basal melt and outflowing heat and salt discharge.”_l. 211-213, but we actually evaluated the buoyancy rather than salt discharge. So, we rectify this inaccurate term “salt discharge” as <buoyancy discharge> therein.

The term ‘salinization’ appears in three places. The first is within the Abstract, that is, “... a significant amplification in channelized basal melting, meltwater channeling, and warming and salinization of the channel flow”_l. 19-21, the second is within the elucidation on the limitation of plume models in the first section “Introduction”, that is, “... and the entrainment-induced warming and salinization, respectively, of the plumes.”_l. 76-77, and the last is in the end of the main text, that is, “... deep channels may correspond to substantial amplification in channelized basal melting, meltwater channeling, and plume warming and salinization (as illustrated in Fig. 9).”_l. 470-472. Therefore, it should be appropriate that the statement, i.e., <These changes in the vertical thermal structure are consistent with that in the vertical haline (**Supplementary Fig. 1**) and thus stratification structures (Supplementary Fig. 2a-d).>_L. 289-291, remains as it is in the current place, i.e., immediately following the discussion on the vertical thermal structure, to keep the contextual coherence. After that, the discussions on the ‘salinization’ can be contextually given in sequence, that is, <That mixing substantially warms (Fig. 5c) and salinizes (**Supplementary Fig. 1b**) the GMW, so the resultant water mass is referred to as intermediate GMW (iGMW) here, ...>_L. 310-312, <..., which makes the water, interacting with the **western-channel** ice base, warmer (Fig. 5c) and saltier (**Supplementary Fig. 1b**) than ...>_L. 323-324, <..., the recirculation of even warmer (Fig. 5d, e) and saltier (**Supplementary Fig. 1c, d**) iGMW upward along the western channel flank ...>_L. 331-333, and <In other words, there might exist a transition into warmer and saltier near-base hydrography if basal channels deepen, ...>_L. 463-465.

In line 447, should Fig.6,d,f be Fig.8 d,f?

Presently, Fig. 7d, f.

Figures:

In Figure 7, the thick arrows intended to represent the GMW or iGMW circulation are misleading as they appear to depict vertical movements like downwelling or upwelling rather than horizontal cross-channel flow. Adjustments to these graphical representations are necessary to accurately reflect the thermodynamic processes being studied.

The top panel of Figure 9, which depicts processes outlined in the manuscript’s earlier version, such as secondary overturning, is not discussed in the current revision. I recommend integrating this panel with Figure 7 to enhance clarity and effectively illustrate these processes.

Thank you for these constructive comments on these figures. Following your

suggestions, we remove the thick arrows in Fig. 7 with the thin arrows vertically distributed with equal intervals to represent the horizontal cross-channel flow, and the internal annotations are correspondingly modified. Then, we integrate the modified Fig. 7 with the top panel of Fig. 9 in which all the elements inside the channels and their annotations are removed to get the new Fig. 8. Please note that the related figure order is also adjusted accordingly.

Figure 8. Schematic representation of channelized ISOBCs. The upper panel shows a comparison of critical physical processes underneath the **deep** (left) and **shallow** (right) basal channels. **Shallow channels** have little influence on the properties of the across-channel GMW flow with little GMW recirculated along the Coriolis-favored flank of channel. In contrast, the stratification and hydrodynamics are significantly changed in deep channels by four critical coherent processes as illustrated, which leads to magnified channelized basal melt and larger meltwater channeling. The arrow colour transitioning from blue to pinkish red denotes the transformation of GMW to warmer and saltier iGMW, and vice versa. The lower panel sketches warm channelized GMW surfacing that leads to the formation of coastal sensible-heat polynyas by melting the fast ice adjacent to an ice shelf, and has a wealth of implications for the multidisciplinary Antarctic oceanography. Not to scale.

Response to Reviewer #3:

In my previous review, my main comments related to the presentation of the paper and some elements of the figure presentation. Looking over the rebuttal & resubmitted manuscript, the authors have addressed these concerns suitably.

We thank the reviewer for their support.

One comment on structure however: the authors have added a section to the discussion relating to model resolution – this is introducing new results, so seems a bit disjointed sitting in the discussion section. I suggest a short section introducing these results is added to the results section, rather than introducing them in the discussion section. The results can then be discussed without disrupting the flow of the discussion.

Thank you for this important comment. Because in the 1st round of review Reviewer #1 recommended moving the discussion on model resolution to the SI, we move the part you mentioned (l. 448-464) in the section “Discussion” to the new Supplementary S1 “Limitation of the current resolution” (L. 57 in SI; the original Supplementary S1 is thus renumbered as S2). Accordingly, we introduce this part in the end of the primary subsection “Importance of basal channel cross-sectional shape in ice-ocean interactions”, that is,

<At last, we conduct six additional runs using higher resolution ($dy=250$ m and/or $dz=2$ m) to examine the reliability of quantified relations between channelized quantities and channel CSS proposed here for the shallowest (60 m) and the narrowest (4 km) cases (see Supplementary S1). It is found that H60W4 has the relatively largest deviation among all these six runs because of the deficiency in representing the smallest cross-sectional area. To this end, an imperative effort needs to be made to extend the applicability of all these empirical expressions of channelized quantities beyond the present range of CSS.>_L. 371-378.

As such, a corresponding statement remains in the section “Discussion” as

<Therefore, the quantification of channelized outflowing mass, heat, and buoyancy discharge of meltwater established here (Fig. 7c, d, f) has the potential to reduce these uncertainties, but it is also worth, as stressed above, exploring the quantified relations between channelized quantities and channel CSS beyond the present CSS range in a later study.>_L.452-457.

REVIEWERS' COMMENTS

Reviewer #1 (Remarks to the Author):

The authors have made the suggested minor adjustments from the 2nd round of reviewers. Therefore, I believe the manuscript is ready for publication.

In an effort to not be overly nitpicky about the flow/word choice in the article, I would also recommend the authors (maybe particularly the UK-based authors?) to do a re-read and make any necessary minor edits in word choice that can help improve the flow and precision of the presentation.

Reviewer #1 (Remarks on code availability):

The code appears to be in a form that is a usable resource for the community.

Reviewer #2 (Remarks to the Author):

The authors have successfully addressed my previous concerns, significantly improving the manuscript's quality. I support its publication in Nature Communications as it now presents a valuable contribution to its field

Reviewer #3 (Remarks to the Author):

I am happy that the authors have addressed the issue I raised in the previous review - moving the text relating to further results out of the discussion section, into the results / supplementary info.

Response to reviewers:

Ice shelf basal channel shape determines channelized ice-ocean interactions

Chen Cheng, Adrian Jenkins, Paul R. Holland, Zhaomin Wang, Jihai Dong, Chengyan Liu

We are grateful to Qin Zhou and other two anonymous reviewers for their great support for the publication of this manuscript. In the following, italic denotes the reviewers' comments, and the following is our response.

Response to Reviewer #1:

The authors have made the suggested minor adjustments from the 2nd round of reviewers. Therefore, I believe the manuscript is ready for publication.

We thank the reviewer for their support.

In an effort to not be overly nitpicky about the flow/word choice in the article, I would also recommend the authors (maybe particularly the UK-based authors?) to do a re-read and make any necessary minor edits in word choice that can help improve the flow and precision of the presentation.

Thank you for this suggestion. As you recommended, the coauthor Adrian Jenkins has done a careful re-read and made the necessary edits in the attached files 'manu2-r3.docx' and 'supplementary material-r3.pdf' to improve the flow and precision of the presentation.

Response to Reviewer #2 (Qin Zhou):

The authors have successfully addressed my previous concerns, significantly improving the manuscript's quality. I support its publication in Nature Communications as it now presents a valuable contribution to its field.

We thank the reviewer for their support.

Response to Reviewer #3:

I am happy that the authors have addressed the issue I raised in the previous review-moving the text relating to further results out of the discussion section, into the results/supplementary info.

We thank the reviewer for their support.